# Well-defined diatomic catalysis for photosynthesis of $C_2H_4$ from $CO_2$

Zhongkai Xie[1], Shengjie Xu[1], Longhua Li[1], Shanhe Gong[1], Xiaojie Wu[1], Dongbo Xu[1], Baodong Mao[1], Ting Zhou[1], Min Chen[1], Xiao Wang [2], Weidong Shi [1]✉ & Shuyan Song [2]✉

Owing to the specific electronic-redistribution and spatial proximity, diatomic catalysts (DACs) have been identified as principal interest for efficient photoconversion of $CO_2$ into $C_2H_4$. However, the predominant bottom-up strategy for DACs synthesis has critically constrained the development of highly ordered DACs due to the random distribution of heteronuclear atoms, which hinders the optimization of catalytic performance and the exploration of actual reaction mechanism. Here, an up-bottom ion-cutting architecture is proposed to fabricate the well-defined DACs, and the superior spatial proximity of CuAu diatomics (DAs) decorated $TiO_2$ (CuAu-DAs-$TiO_2$) is successfully constructed due to the compact heteroatomic spacing (2-3 Å). Owing to the profoundly low C-C coupling energy barrier of CuAu-DAs-$TiO_2$, a considerable $C_2H_4$ production with superior sustainability is achieved. Our discovery inspires a novel up-bottom strategy for the fabrication of well-defined DACs to motivate optimization of catalytic performance and distinct deduction of heteroatom synergistically catalytic mechanism.

Photocatalytic carbon dioxide ($CO_2$) reduction, a mimicking natural photosynthesis, is identified as an ideal technology to reduce the $CO_2$ level by the conversion of $CO_2$ into fuels or industrial feedstocks with the utilization of solar energy[1–3]. Among all the photoreduction $CO_2$ products, ethylene ($C_2H_4$) is considered as high-value species due to their high energy densities and commercial prices in chemical industry[4–13]. Bimetallic solid-photocatalysts, consisted of ordered stagger of heteronuclear metal atoms, can strikingly reduce C-C coupling energy barrier via weakening dipole-dipole repulsion of the neighboring adsorbed C-based intermediates, which has been regarded as the dominator in photosynthesis of $C_2H_4$[14–21]. Nevertheless, the compact atom stacking during the synthesis of bimetallic solid catalysts inevitably contributes to the decrease of active sites[22–27], critically restricting the efficient $C_2H_4$ generation. Therefore, its deep desirability to accurately optimize bimetallic solid catalysts at the atomic scale to break the bottleneck of inherent scaling-catalytic relationship in photosynthesis of $C_2H_4$.

Single-atom catalysts (SACs), a well-defined mononuclear metal sites, which have attracted huge attention for their potential to overcome the disadvantages of previously developed solid catalysts[27]. Conspicuously, heteronuclear DACs can not only maintain the maximized atom utilization but also modulate the reaction kinetics and even the reaction pathways by involving two metal atoms with cooperative modification of their steric and electronic properties[26–31]. So far, all reported heteronuclear DACs fabrication have concentrated on bottom-up synthetic strategies, such as organometallic compounds pyrolysis[26,32–35], metal complexation[36–38], metal coprecipitation[39–43], in situ photoreduction[44,45], and physical desorption[46,47]. Because of the repellency between the bare metal atoms under such prevailing bottom-up synthesis methods, disordered heteronuclear sites distribution have been an insurmountable barrier, which directly restrained the optimization of catalytic performance and distinct deduction of catalytic mechanisms[48]. Consequently, it is of high priority to exploit novel strategy for well-defined heteronuclear DACs

[1]School of Chemistry and Chemical Engineering, Jiangsu University, Zhenjiang 212013, China. [2]State Key Laboratory of Rare Earth Resource Utilization, Changchun Institute of Applied Chemistry, Chinese Academy of Sciences, Changchun 130022, China. ✉e-mail: swd1978@ujs.edu.cn; songsy@ciac.ac.cn

fabrication to establish a high-efficiency photosynthesis of $C_2H_4$ system.

Here, an up-bottom ion-cutting engineering for the synthesis of atomic-level catalysts was initially proposed. Depending on the vectored etching of Cu in a CuAu alloy (isolated Au atoms in the Cu lattice), the CuAu-DAs with subnanometer heteroatomic spacing supported by commercial $TiO_2$ (CuAu-DAs-$TiO_2$) were successfully fabricated. The $C_2H_4$ production of CuAu-DAs-$TiO_2$ proceeded at a remarkable rate of 568.8 $\mu mol \cdot g^{-1} \cdot h^{-1}$ without any sacrificial agent, which is superior to recent reported works in photoconversion $CO_2$ and $H_2O$ into $C_2H_4$ (Supplementary Table 1), and no apparent catalyst deactivation was observed during the 120-h photocatalytic stability test. In such CuAu-DAs structures, Cu single atoms (Cu-SAs) are mainly responsible for the high-efficiency *CO generation rather than C-C coupling centers[16,17,49–52], while Au single atoms (Au-SAs) serve as *CO coupling centers to rapidly consume the *CO arising from Cu-SAs according to photocatalytic $CO_2$ reduction, which synergistically promotes the high efficiency and sustainability of photoconversion of $CO_2$ into $C_2H_4$[53].

## Results

### Catalyst synthesis and structural characterization

Compared to the widespread bottom-up synthetic strategies of SACs and DACs[32–48,50–58], the up-bottom ion-cutting architecture controlled by the vectored etching of specific element contents in the alloy may provide novel insight for the adjustable design of SACs and DACs.

Figure 1a illustrates the ion-cutting architecture fabrication of Cu nanoclusters (Cu-NCs) decorated Au-SAs (CuAu-NCSAs), CuAu-DAs, and Au-SAs/Cu-SAs supported by $TiO_2$ via adjusting the vectored etching time of the $Cu_5Au_1$ alloy. Transmission electron microscopy (TEM) and aberration-correction high-angle annular dark-field scanning transmission electron microscopy (AC-HAADF-STEM) were carried out to acquire a more spatially resolved structural configuration during the different catalyst synthesis processes. In Supplementary Figs. 1 and 2, the commercial $TiO_2$ displays the irregular nanoparticle morphology with the ca. 13.6 nm of particle size, and the EDS mapping analysis also verifies the uniform Ti and O elements distribution. Compared to commercial $TiO_2$, the regular and larger size of $Cu_5Au_1$ alloy was observed in Supplementary Fig. 3a, and the compactly connected heterointerface was observed between $Cu_5Au_1$ and $TiO_2$ (Supplementary Fig. 3b). In addition, the overlapping Cu and Au elements distribution were also exhibited in EDS mapping images, indicating the successful construction of CuAu alloy (Supplementary Fig. 3c–g). As shown in Fig. 1b and Supplementary Fig. 4, the ordered stagger of Cu/Au atoms stacked $Cu_5Au_1$ alloy nanoparticles are successfully loaded on bulk commercial $TiO_2$, and the isolated Au single atom in Cu lattice provided the precondition for CuAu-DAs formation. Moreover, the 4.43 and 1.71 Å of Au-Au and Cu-Au atomic distance further verify the isolated Au SAs surrounded by Cu atoms rather than the Au atoms in $Cu_5Au_1$ alloy (Supplementary Fig. 4a–c). After 1 h of vectored etching of Cu atoms (Fig. 1c and Supplementary Fig. 5a–d), slightly destroyed lattice and few Cu vacancies in the $Cu_5Au_1$ alloy were observed due to

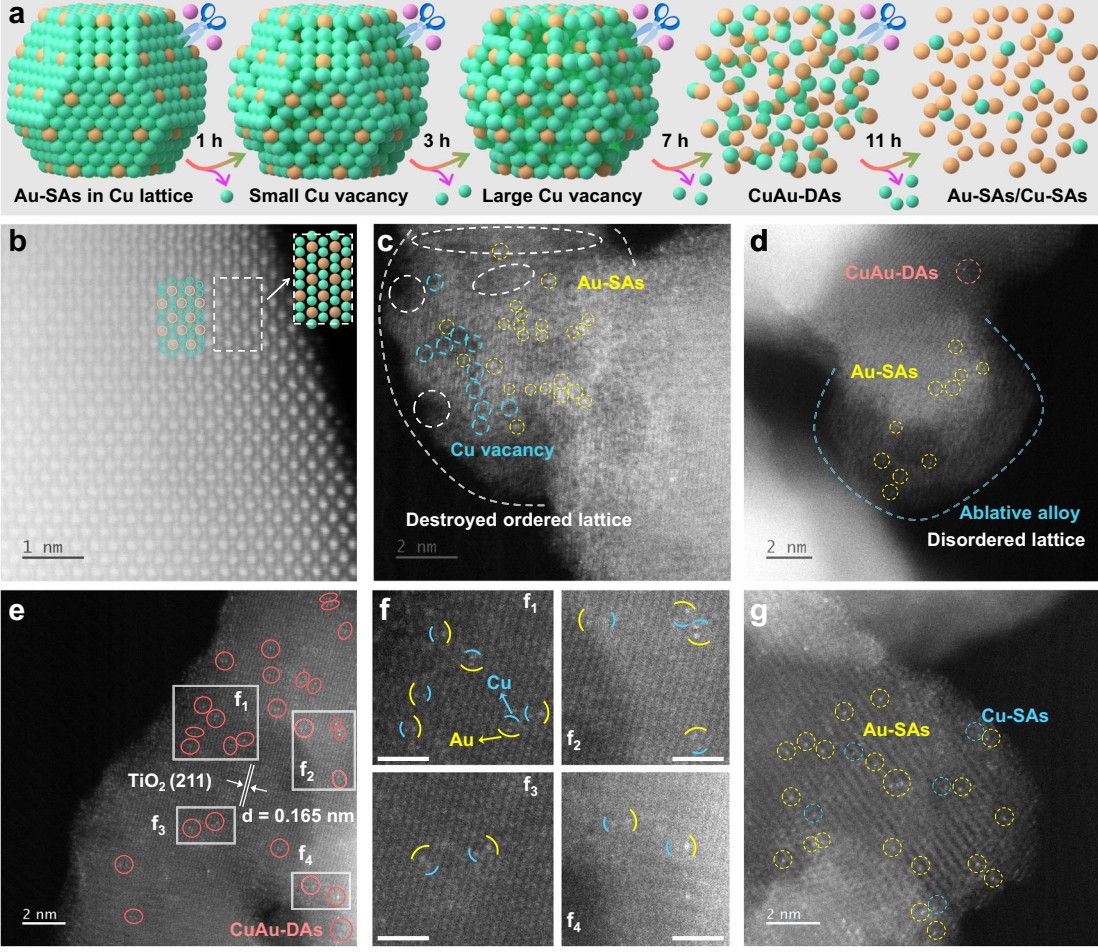

**Fig. 1 | Morphological characterization of the CuAu-based $TiO_2$ composites.**
**a** Dynamic schematic illustration from isolated Au-SAs in the Cu lattice to CuAu-DAs during the vectored etching process (green sphere: $Cu^0$; orange sphere: $Au^0$; purple sphere: $Fe^{3+}$). AC-HAADF-STEM images of $Cu_5Au_1$-$TiO_2$ (**b**), $E_1$-$Cu_5Au_1$-$TiO_2$ (**c**), $E_3$-$Cu_5Au_1$-$TiO_2$ (**d**), $E_7$-$Cu_5Au_1$-$TiO_2$ (**e**), and $E_{11}$-$Cu_5Au_1$-$TiO_2$ (**g**). **f** Magnified images of the corresponding areas in image (**e**). Scale bars of $f_1$-$f_4$: 1 nm.

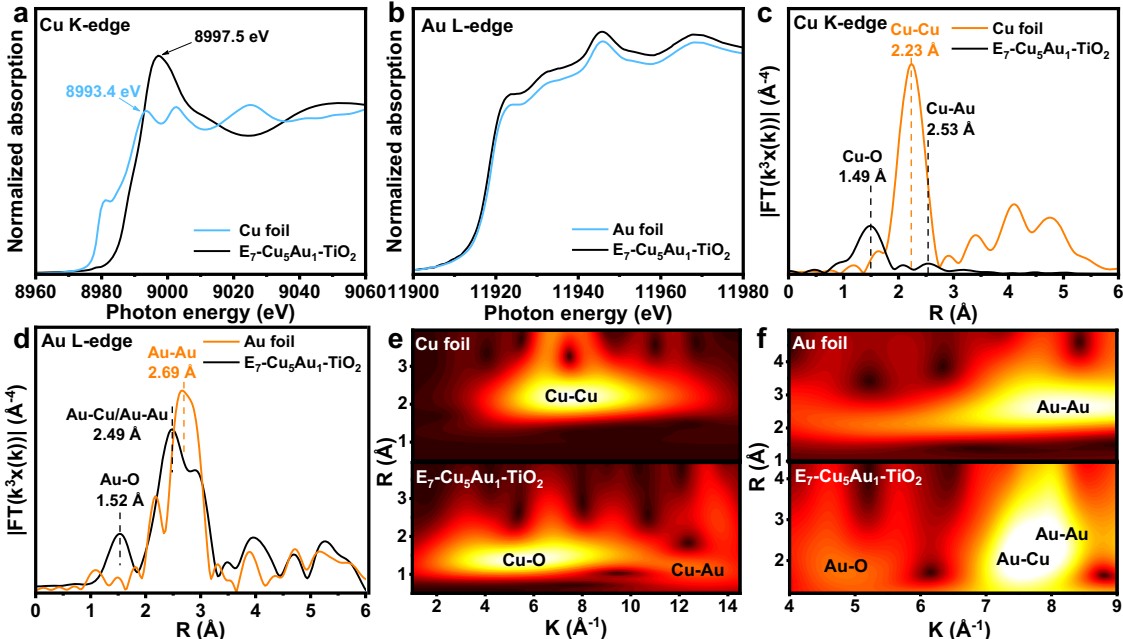

**Fig. 2 | Structural characterization of the CuAu-DAs.** XANES analysis of E$_7$-Cu$_5$Au$_1$-TiO$_2$ and reference samples at the Cu K-edge (**a**) and Au L-edge (**b**). Corresponding k$^3$-weighted FT-EXAFS spectra in the R space for E$_7$-Cu$_5$Au$_1$-TiO$_2$ and references at the Cu K-edge (**c**) and Au L-edge (**d**). **e** Cu K-edge WT-EXAFS spectra of Cu foil and E$_7$-Cu$_5$Au$_1$-TiO$_2$. **f** Au L-edge WT-EXAFS spectra of Au foil and E$_7$-Cu$_5$Au$_1$-TiO$_2$.

the destruction of the ordered arrangement of Cu-Au atoms. The ablative Cu$_5$Au$_1$ alloy and few CuAu-DAs were obtained after 3 h of etching (Fig. 1d and Supplementary Fig. 6a, b), indicating that the constant loss of Cu atoms could promote the collapse of the Cu$_5$Au$_1$ alloy framework and the redistribution of adjacent Cu-Au atoms on the TiO$_2$ surface. Moreover, the distinguished ordered lattice of TiO$_2$ and disordered lattice of Cu$_5$Au$_1$ were simultaneously observed in HRTEM and the corresponding pseudo-color images, which further implied the successfully vectored etching of such Cu$_5$Au$_1$ alloy (Supplementary Fig. 6c, d). As shown in Fig. 1e, the CuAu-DAs are uniformly dispersed on the surface of the TiO$_2$ (211) plane (lattice spacing: 0.165 nm) in E$_7$-Cu$_5$Au$_1$-TiO$_2$. In addition, the magnified images (Fig. 1f$_1$-f$_4$) and acquired AC-HAADF-STEM image intensity profile (Supplementary Fig. 7) clearly verified subnanometer distances (~2–3 Å) between the Cu-SAs (dark) and Au-SAs (bright) according to the different atomic mass of the corresponding elements[59], suggesting that the twin CuAu-DAs was successfully constructed after 7 h of vectored etching of the Cu$_5$Au$_1$ alloy. Moreover, when the vectored etching time was prolonged to 11 h (Fig. 1g and Supplementary Fig. 8), a few Cu-SAs and dominated Au-SAs were observed in the AC-HAADF-STEM image, indicating that the dynamic vectored etching process promoted the further dissociation of CuAu-DAs structure due to the consistent decrease of Cu atoms. In Supplementary Fig. 9, Cu-SAs modified TiO$_2$ was also constructed by 7 h of vectored etching of Cu-TiO$_2$, suggesting the huge potential of such up-bottom ion-cutting technology for atomic-level catalysts design.

To further investigate the atomic-scale configuration of Cu and Au in E$_7$-Cu$_5$Au$_1$-TiO$_2$, Cu K-edge and Au L-edge X-ray absorption near-edge structure (XANES) spectroscopy was performed (Fig. 2a, b). Figure 2a illustrates the Cu K-edge XANES spectra for E$_7$-Cu$_5$Au$_1$-TiO$_2$ with Cu foil benchmarks as reference, and the absorption edge position of E$_7$-Cu$_5$Au$_1$-TiO$_2$ is more positive than that of Cu foil, indicating that partial Cu could directly connect with the lattice O of TiO$_2$. In Fig. 2b, the white line peak of E$_7$-Cu$_5$Au$_1$-TiO$_2$ exhibits Au characteristic features, which is similar to those of the reference Au foil, indicating the presence of Au$^0$. As shown in Fig. 2c, the local coordination around the Cu-O shell and Cu-M shell (Cu-Cu and Cu-Au) was determined by

the k$^3$-weighted Fourier transform of the extended X-ray absorption fine structure (FT-EXAFS) spectrum. A predominant peak at ~1.49 Å of E$_7$-Cu$_5$Au$_1$-TiO$_2$ is assigned to the Cu-O coordination. Furthermore, the other obvious characteristic peak at ~2.53 Å is observed in the E$_7$-Cu$_5$Au$_1$-TiO$_2$ spectrum but not in the Cu foil spectrum, implying the possible formation of Cu-Au coordination. The local peak at ~1.52 Å in the Au L-edge spectrum of CuAu-DAs-TiO$_2$ (Fig. 2d) is close to that of Au-O, indicating the formation of Au-O coordination. An apparent path at ~2.49 Å is observed in the Au L-edge spectrum of E$_7$-Cu$_5$Au$_1$-TiO$_2$ (Fig. 2d), which is close to the values of Cu-Au and Au-Au (Fig. 2c). The Cu foil is used to calculate the standard amplitude reduction factor ($S_0^2 = 0.845$, Supplementary Table 2), and the Cu K-edge EXAFS analysis of E$_7$-Cu$_5$Au$_1$-TiO$_2$ in R spaces is exhibited in Supplementary Fig. 10. The EXAFS spectrum of E$_7$-Cu$_5$Au$_1$-TiO$_2$ is analyzed by using two backscattering paths (Cu-O and Cu-Au). The best-fitting results exhibited that the coordination number of the O and Au in the first coordination sphere of E$_7$-Cu$_5$Au$_1$-TiO$_2$ is fitted to be ≈3.3 and ≈1.2 at distances of 1.93 and 2.91 Å, respectively, implying the Cu-SAs is merely adjacent with single Au atom. Therefore, there is no doubt that the existence of CuAu-DAs structure under such vectored etching process, while it is still recognized indeed small existence of pure Au phase due to the shortage of such solvothermal method for the fabrication of highly ordered CuAu alloy. Moreover, the concurrent detection of Cu-M and Au-M distances is in the range of the observed distribution of twin dual atoms in atomic-resolution STEM imaging (Supplementary Fig. 7), further demonstrating that the Cu-Au bond originated from the initial CuAu alloy rather than from Cu-SAs and Au-SAs rearrangement during the etching process. The Cu K-edge and Au L-edge EXAFS oscillations are also analyzed by the wavelet transform (WT) method to further confirm the presence of the Cu-Au path. No WT maxima at 7.5 Å$^{-1}$ (Cu-Cu bond) and 8.3 Å$^{-1}$ (Au-Au bond) were observed in the spectra of E$_7$-Cu$_5$Au$_1$-TiO$_2$ (Fig. 2e, f), indicating the SAs-structure configuration of most Cu and Au sites. The WT maxima at 5.7 Å$^{-1}$ and 13 Å$^{-1}$ in the spectra of E$_7$-Cu$_5$Au$_1$-TiO$_2$ correspond to the Cu-O and Cu-Au bonds, respectively (Fig. 2e). Similarly, the WT maxima at 4.9 Å$^{-1}$ and 7.7 Å$^{-1}$ are attributed to the Au-O and Au-Cu/Au-Au bonds (Fig. 2f), respectively, consistent with the FT-EXAFS results (Fig. 2c, d).

Consequently, the adjacent Cu and Au coexisted as twin diatomic centers and connected with the lattice O of $TiO_2$ to form O-Cu-Au-O.

As shown in Supplementary Fig. 11a, the Cu 2p X-ray photoelectron spectroscopy (XPS) binding energy peaks at approximately 932.23 and 952.03 eV are attributed to $Cu^0/Cu^+$[60]. The Auger electron spectrum (Cu LMM) was obtained to further distinguish between $Cu^0$ and $Cu^+$ in $Cu$-$TiO_2$ and $Cu_5Au_1$-$TiO_2$. As shown in Supplementary Fig. 12, the characteristic kinetic energy peaks at 918.06 and 921.82 eV[61] have corresponded to the electron state of $Cu^0$ in $Cu$-$TiO_2$ and $Cu_5Au_1$-$TiO_2$. The presence of weak peaks located at 934.70 and 954.40 eV confirms the presence of trace $Cu^{2+}$ in $Cu$-$TiO_2$ and $Cu_5Au_1$-$TiO_2$. Notably, no $Cu^0$ or $Cu^{2+}$ were detected in $E_7$-$Cu_5Au_1$-$TiO_2$ because Cu was below the detection limits after the cooperative $Fe^{3+}$ and $H^+$ etching reaction of $Cu^0$ and $Cu^{2+}$, consistent with the rare residual Cu-SAs observed in AC-HAADF-STEM (Fig. 1e). In Supplementary Fig. 11b, the two Au 4f peaks of Au-$TiO_2$ located at 82.99 (Au $4f_{7/2}$) and 86.70 eV (Au $4f_{5/2}$) are attributed to zero valence Au. Compared to those of Au-$TiO_2$, the Au $4f_{7/2}$ and Au $4f_{5/2}$ characteristic peaks of $Au_5Cu_1$-$TiO_2$ are positively shifted to 83.18 and 86.90 eV, indicating the impeded electron transfer from $TiO_2$ to Au due to the surrounding Cu barrier. After the etching process, the Au 4f binding energy peaks of $E_7$-$Cu_5Au_1$-$TiO_2$ are more negative than those of Au-$TiO_2$ and $Cu_5Au_1$-$TiO_2$, illustrating that more $TiO_2$ local electrons are transferred to Au due to the decreased amounts of Cu, further suggesting the direct concatenation between Au and Cu. The most negative Au 4f binding energies of $E_7$-$Cu_5Au_1$-$TiO_2$ suggest the high-concentration electron density of Au, which inevitably benefits the multiple electron reaction conduction on the Au sites. Consequently, large amounts of photogenerated electrons may be transferred from $TiO_2$ to the CuAu-DAs due to the low Femi level of metallic Cu and Au[62,63], which may cause both Cu-SAs and Au-SAs to be the main centers for $CO_2$ adsorption-activation and C-C coupling. Significantly, the Ti 2p spectra of $E_7$-$Cu_5Au_1$-$TiO_2$ exhibits peaks of Ti $2p_{3/2}$ and Ti $2p_{1/2}$ at more negative binding energies compared to those of $Cu$-$TiO_2$, Au-$TiO_2$, and $Cu_5Au_1$-$TiO_2$ (Supplementary Fig. 11c), further indicating that the Cu-SAs and Au-SAs are more beneficial for photogenerated electron migration from $TiO_2$ to the CuAu-DAs under irradiation. All the O 1s spectra of the as-prepared samples show two typical peaks, which are assigned to the O-Ti bond of $TiO_2$ and the O−H bond of surface adsorbed OH groups[62] (Supplementary Fig. 11d). X-ray diffraction (XRD) was conducted to further analyze the crystal surface information of these $TiO_2$-based samples. Compared to anatase $TiO_2$ (JCPDS No. 71-1166), the as-prepared Au-$TiO_2$ and $Cu_5Au_1$-$TiO_2$ composites gives rise to (200), (220), and (311) characteristic peaks of Au (JCPDS 65-2870), while no characteristic peak of Cu was observed in $Cu_5Au_1$-$TiO_2$ and $Cu$-$TiO_2$ (Supplementary Fig. 13), indicating the rapid growth of pure Au phase and the restrained ordered growth of Cu in the CuAu structure due to the different standard electrode potential ($E^0$) of Au ($[AuCl_4]/Au^0$, $E^0 = +0.93$ V) and Cu ($Cu^{2+}/Cu^0$, $E^0 = +0.34$ V)[64]. Notably, the HRTEM and EDS mapping images exhibited the small-sized Cu nanoparticle (ca. 7.9 nm) decorated $TiO_2$ (Supplementary Fig. 14a−d), and the EDS mapping also showed the uniformly dispersed Cu elements (Supplementary Figs. 14 and 15), which indicated that Cu existed as the small-sized scale rather than extended growth into large-sized structure (Supplementary Fig. 14e−h). There has been reported that the low ordering degree of CuAu alloy merely display the Au characteristic peaks due to the dominated Miller indexes of Au phase[65], which is corresponded to the observation of weak Au characteristic peaks in our work. Although the AC-HAADF-STEM and EDS mapping display the Cu atoms surrounded isolated Au single atoms (Supplementary Fig. 4d−g), the predominated 0.233 nm of lattice spacing was observed in the AC-HAADF-STEM of $Cu_5Au_1$-$TiO_2$, which is ascribed to the (111) crystal facet of Au rather than Cu and CuAu characteristic lattice planes. Therefore, the apparent characteristic crystal facet of Au that displayed in AC-HAADF-STEM images and XRD reflections could be ascribed to the dominated Miller indexes of Au phase (Supplementary Figs. 4d and 13). According to the dynamic XRD patterns of $Cu_5Au_1$-$TiO_2$ after different vectored etching time (Supplementary Fig. 16a, b), no obvious enhanced Au characteristic peak intensity of $E_t$-$Cu_5Au_1$-$TiO_2$ was observed compared to that of $Cu_5Au_1$-$TiO_2$ after the vectored etching process, suggesting that the vectored etching processes of Cu could not promote isolated Au-SAs rearrangement to form Au lattice plane. Inductively coupled plasma-atomic emission spectroscopy (ICP–AES) was conducted to evaluate the dynamic Cu/Au molar ratios during the etching process. In Supplementary Fig. 17, with increasing etching time, the Cu/Au molar ratio value of $E_t$-$Cu_5Au_1$-$TiO_2$ decreases deeply, while the molar ratio values of $H_t$-$Cu_5Au_1$-$TiO_2$ decrease slightly, indicating the presence of large amounts of $Cu^0$ rather than $Cu^+/Cu^{2+}$ and the successful construction of an adjustable CuAu atomic-level nanostructure. Owing to the different redox capacity-related standard electrode potential between $Fe^{3+}$ ($Fe^{3+} + e^- \rightarrow Fe^{2+}$, $E^0 = +0.77$ V) and $Cu^0$ ($Cu^{2+} + 2e^- \rightarrow Cu^0$, $E^0 = +0.34$ V), the $Fe^{3+}$ could spontaneously be reduced by $Cu^0$ ($Fe^{3+} + Cu^0 \rightarrow Fe^{2+} + Cu^{2+}$)[64]. However, due to the insufficient oxidized capacity of $Fe^{3+}$ for $Au^0$ ($Au^{3+} + 3e^- \rightarrow Au^0$, $E^0 = +1.52$ V) oxidation[64], the contents of Au keep constant during the whole etching process (Supplementary Table 3), which seriously restrains the dissolution of $Au^0$ and promote the redistribution of Au atoms on $TiO_2$. Localized surface plasmon resonance (LSPR) is often related to the metal shape and the dielectric constant of the surrounding medium[66]. Therefore, the connection between atomic interface engineering and the light absorption of the as-prepared samples was also checked by UV–vis–NIR diffuse reflection spectroscopy (DRS). As shown in Supplementary Fig. 18a, an obvious redshift is observed over metal-decorated $TiO_2$ compared to pure $TiO_2$, implying that Cu and Au could effectively enhance visible light absorption and produce more photogenerated carriers. In Supplementary Fig. 19a, $E_t$-$Cu_5Au_1$-$TiO_2$ shows strikingly increased Au (~520 nm) and Cu (~730 nm) LSPR response peaks compared to $Cu_5Au_1$-$TiO_2$, indicating that the regularly variational nanoscale of Au and Cu can benefit the formation of higher electron density centers upon irradiation. However, with the persistent etching of Cu, the LSPR response of Cu almost disappeared due to the consistently decreased content of Cu. Moreover, $H_t$-$Cu_5Au_1$-$TiO_2$ show much weaker Au LSPR response than $E_t$-$Cu_5Au_1$-$TiO_2$ (Supplementary Fig. 19b, c), and the negligible Cu LSPR response intensity of $H_t$-$Cu_5Au_1$-$TiO_2$ confirms that the $H^+$ etching process eliminated only a small amount of surficial oxidized Cu to a certain extent, which implied that $E_t$-$Cu_5Au_1$-$TiO_2$ could induce a higher photogenerated carrier density on CuAu sites due to the stronger Cu and Au LSPR response intensity.

To reveal the charge carrier dynamics on CuAu-DAs modified $TiO_2$, photoluminescence (PL) spectroscopy and time-resolved photoluminescence (TRPL) spectroscopy were carried out. As shown in Supplementary Fig. 20a, the intensity of the emission peaks of $Cu$-$TiO_2$, $Cu_5Au_1$-$TiO_2$, and $E_7$-$Cu_5Au_1$-$TiO_2$ at ~425 nm decrease considerably compared to that of pure $TiO_2$, indicating that the decoration of metallic cocatalysts on $TiO_2$ can improve the charge separation and transfer. Notably, the spectrum of $E_7$-$Cu_5Au_1$-$TiO_2$ displays a weaker emission peak than that of $Cu_5Au_1$-$TiO_2$, indicating the lower photogenerated charge recombination of $E_7$-$Cu_5Au_1$-$TiO_2$ due to the feedthrough charge transfer channel between $TiO_2$ and CuAu-DAs. It is reported that the contributions of $\tau_1$ and $\tau_2$ are more related to charge transfer, and the PL decay is more dominated by $\tau_3$[67–69]. In Supplementary Fig. 20b and Table 4, the CuAu-DAs modified $TiO_2$ exhibits shorter $\tau_1$, $\tau_2$, and $\tau_3$ compared to $TiO_2$, illustrating the compact interaction and suppressed charge recombination between CuAu-DAs and $TiO_2$. Moreover, the shortest $\tau_3$ lifetime of CuAu-DAs modified $TiO_2$ represents the fastest decay in CuAu-DAs-$TiO_2$, which is ascribed to the fact that the direct connection between CuAu-DAs and $TiO_2$ is more beneficial for convenient photogenerated charge transfer from $TiO_2$ to CuAu-DAs rather than recombination in the Cu and $Cu_5Au_1$ bulk. Transient photocurrent and electrochemical impedance

spectroscopy (EIS) measurements were also conducted to further reveal the efficiency of photogenerated charge separation and transportation of the as-synthesized samples. In Supplementary Fig. 20c, d, the highest photocurrent density and smallest arc radius of $E_7$-$Cu_5Au_1$-$TiO_2$ suggest that the CuAu-DAs exhibits much higher charge separation efficiency and faster interfacial charge transportation than $TiO_2$, Cu-$TiO_2$, and $Cu_5Au_1$-$TiO_2$. Therefore, the superior photoreduction $CO_2$ performance of CuAu-DAs-$TiO_2$ could also be attributed to the excellent photoabsorption and high-efficiency charge separation.

## Photocatalytic performance toward $CO_2$ photoreduction

A suitable band edge position of $TiO_2$ is a prerequisite for the successful photoconversion of $CO_2$ gas and $H_2O$ vapor into CO, $CH_4$, $C_2H_4$, $C_2H_6$, and $O_2$ upon 320–780 nm irradiation (Supplementary Fig. 18e). To uncover the influence of different CuAu atomic interfaces on the $CO_2$ photoreduction reaction, photocatalytic $CO_2$ reduction tests were carried out on different CuAu nanostructure-decorated $TiO_2$ samples. At the same loading amounts, Cu-$TiO_2$ and Au-$TiO_2$ achieved much higher CO and $CH_4$ production rates than pure $TiO_2$ (Fig. 3a and Supplementary Fig. 21a, b) due to the increased number of active sites and the improved charge transfer efficiency[70]. As shown in Fig. 3a and Supplementary Fig. 21a, b, $Cu_5Au_1$-$TiO_2$ performs a higher $CH_4$ production rate (20.4 μmol·g$^{-1}$·h$^{-1}$) than Cu-$TiO_2$ and Au-$TiO_2$, which is attributed to the fact that the Au matrix often functions as an electron sink, while the adjacent surficial Cu serves as $CO_2$ activation centers and proton transfer stations[70,71], and the intermediate further combines with the surrounding abundant electrons originating from the Au matrix to promote $CH_4$ production through the carbene pathway[62,72,73]. The actual active sites merely exist on the exposed alloy surface rather than in the interior of the bulk, resulting in the catalytic reaction typically occurring on the unsaturated active sites of the CuAu alloy surface. Therefore, photocatalytic $CO_2$ reduction tests of the etched samples were also performed to further investigate the influence of CuAu interfacial engineering on the photoreduction of $CO_2$. In Supplementary Fig. 22, a possible schematic illustration is shown to describe the dynamic change in the CuAu structural configuration

during the constant vectored etching process and define a possible correlation with the photocatalytic performance. With increasing vectored etching time, a peculiar dual volcanic relationship of the hydrocarbon production rate is observed in the $E_t$-$Cu_5Au_1$-$TiO_2$ series (Fig. 3b). After 1 h of vectored etching of $Cu_5Au_1$-$TiO_2$, $E_1$-$Cu_5Au_1$-$TiO_2$ reaches the highest $CH_4$ production (68.2 μmol·g$^{-1}$·h$^{-1}$) and electron-based selectivity (84.7%) among the series of $E_t$-$Cu_5Au_1$-$TiO_2$ (Fig. 3b and Supplementary Fig. 21c, d), which is ascribed to the increased number of unsaturated Au and Cu active sites (Supplementary Fig. 22). Nevertheless, the $CH_4$ production rate continuously decrease due to the constant decrease in the number of Cu atom active sites after a longer vectored etching time. Notably, when the etching time is prolonged to 7 h (Fig. 3b, c and Supplementary Fig. 21c, d), the $C_2H_4$ and $C_2H_6$ production rates of $E_7$-$Cu_5Au_1$-$TiO_2$ reach 71.6 and 8.5 μmol·g$^{-1}$·h$^{-1}$, respectively, and the electron-based selectivity reached 68.3% ($C_2H_4$) and 9.4% ($C_2H_6$). The dramatically increased $C_2H_4$ production rate of $E_7$-$Cu_5Au_1$-$TiO_2$ is 305 and 73 times higher than that of $TiO_2$ and $Cu_5Au_1$-$TiO_2$, respectively, which is ascribed to the superior synergistically enhanced C-C coupling over CuAu-DAs. With the further etching of Cu-SAs, the $C_2H_4$ production rate of $E_9$-$Cu_5Au_1$-$TiO_2$ and $E_{11}$-$Cu_5Au_1$-$TiO_2$ constantly decrease due to the constantly decreasing amounts of Cu-SAs, implying that the mere presence of Au-SAs (Fig. 1g) could not satisfy the high-efficiency C-C coupling and $C_2H_4$ production. Moreover, the optimized amount of $E_7$-$Cu_5Au_1$-$TiO_2$ photocatalysts was verified to be 20 mg with a highest $C_2H_4$ production rate based on the excellent mass transfer and superior light utilization (Supplementary Fig. 23), and the apparent decrease of C-based products production rate was observed with the addition of superfluous photocatalysts, which could be ascribed to the impeded light transmission[74]. Compared to Au-SAs modified $TiO_2$, Cu-SAs (Supplementary Fig. 9) modified $TiO_2$ ($E_7$-Cu-$TiO_2$) exhibits dramatically increased CO and $CH_4$ generation, while negligible $C_2$ generation is also observed on $E_7$-Cu-$TiO_2$, indicating that the Cu-SAs is beneficial for high-efficiency *CO production (Supplementary Fig. 24), consistent with recent research results on Cu-SAs based photocatalytic $CO_2$ reduction[75–79]. Therefore, we suspect that only the specific existence of adjacent Cu-SAs and

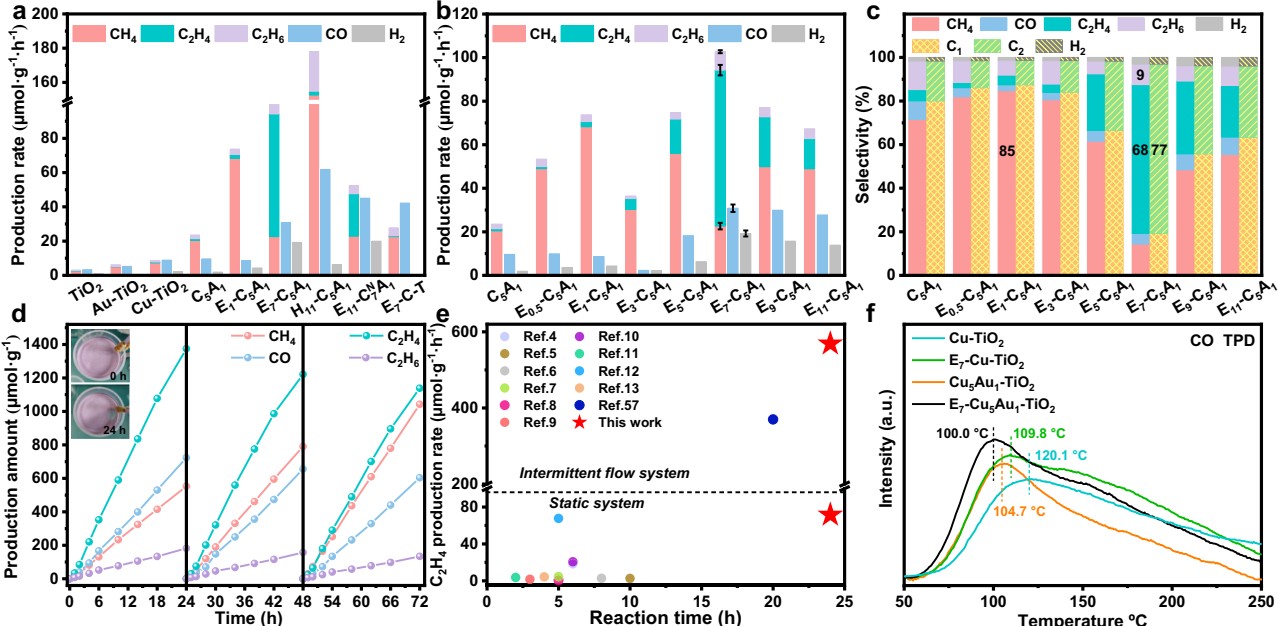

**Fig. 3 | Photocatalytic performance of CuAu-based TiO₂ composites.** $CH_4$, $C_2H_4$, $C_2H_6$, CO, and $H_2$ production rates of the as-prepared photocatalysts (**a**) and $E_t$-$Cu_5Au_1$-$TiO_2$ (**b**). **c** Electron-based selectivity of photocatalytic $CO_2$ conversion over $E_t$-$Cu_5Au_1$-$TiO_2$. **d** Long-term photocatalytic stability test of $E_7$-$Cu_5Au_1$-$TiO_2$. **e** $C_2H_4$ production rate of $E_7$-$Cu_5Au_1$-$TiO_2$ with the reaction time in comparison with recent reports during the closed glass photoreduction of $CO_2$ with a $H_2O$ gas-circulation system without any sacrificial agents. **f** CO-TPD profile of the as-prepared samples.

Au-SAs can efficiently and synergistically favor C-C coupling and $C_2H_4$ production. To further confirm that the CuAu-DAs structure can promote $C_2H_4$ generation, a photoreduction $CO_2$ test was also conducted on $E_t$-$Cu_7^NAu_1$-$TiO_2$ (Supplementary Fig. 25) constructed by adding additional Cu content to the $Cu_5Au_1$ alloy. The dual volcanic relationship of the hydrocarbon production rate is still observed on $E_t$-$Cu_7^NAu_1$-$TiO_2$ based on the transformation of the CuAu structure. Furthermore, $E_{11}$-$Cu_7^NAu_1$-$TiO_2$ exhibits the maximum $C_2H_4$ production rate after 11 h of vectored etching (Supplementary Fig. 25a), and the CO production also rapidly increase with enhanced $C_2H_4$ generation, similar to the behavior of $E_7$-$Cu_5Au_1$-$TiO_2$, indicating that the *CO species is a critical intermediate for C-C coupling, as reported[6,14,53]. As shown in the ICP–AES results (Supplementary Fig. 26 and Table 5), the Cu/Au molar ratios of both $E_7$-$Cu_5Au_1$-$TiO_2$ and $E_{11}$-$Cu_7^NAu_1$-$TiO_2$ are similar, further confirming that the dominated CuAu DAs-structure dramatically actuate $C_2H_4$ production. Moreover, photoreduction $CO_2$ tests were also conducted on $H_t$-$Cu_5Au_1$-$TiO_2$ to further identify the effects of different CuAu structures. Although a superior CO production rate ($61.8\,\mu mol\cdot g^{-1}\cdot h^{-1}$) is observed on $H_{11}$-$Cu_5Au_1$-$TiO_2$ (Supplementary Fig. 27), the $C_2$ production is still nonideal, emphasizing that only Au-SAs, rather than low-coordination Au sites, could rapidly convert the high-concentration CO into $C_2$ products. Furthermore, a superhigh $CH_4$ ($152.6\,\mu mol\cdot g^{-1}\cdot h^{-1}$) production rate is observed for $H_{11}$-$Cu_5Au_1$-$TiO_2$, indicating that low-coordination Au sites in the Cu lattice are more beneficial for converting *CO into $CH_4$ than *CO coupling on Au-SAs. Consequently, in such CuAu-DAs modified $TiO_2$ photocatalytic systems, the Cu-SAs guarantee rapid *CO generation and high-concentration coverage, and the adjacent Au-SAs further promote migration and coupling of the generated *CO. The source of photoreduced $CO_2$ products was investigated by using isotope labeling $^{13}CO_2$ and $H_2^{18}O$ as the reactant under irradiation, and the products were analyzed by gas chromatography–mass spectrometry (GC–MS). The GC–MS peak sequences of $^{13}CO$, $^{13}CH_4$, $^{13}C_2H_4$, and $^{13}C_2H_6$ are shown in Supplementary Fig. 28, and the peaks at $m/z = 29$, $m/z = 17$, $m/z = 30$, and $m/z = 32$ are assigned to $^{13}CO$, $^{13}CH_4$, $^{13}C_2H_4$, and $^{13}C_2H_6$, indicating that the carbon source of CO and hydrocarbons is indeed derived from the input $CO_2$ gas. Overall $CO_2$ photoreduction is divided into two major half reaction steps, the $CO_2$ reduction and $H_2O$ oxidation[12,80], and the detection of $^{16}O^{18}O$ and $^{18}O_2$ species verifies that $O_2$ originates from $H_2O$ oxidation in the photocatalytic $CO_2$ reduction in (Supplementary Fig. 28). Notably, in Supplementary Fig. 29, $O_2$ evolution related to holes consumption of the $TiO_2$ based composites are also stoichiometrically approximate to products of photogenerated electrons reduction, which indicates the simultaneous $CO_2$ reduction and $H_2O$ oxidation behaviors. The $CO_2$ photoreduction experiment was also taken under no existence of $H_2O$ to further figure out the influence of $H_2O$ species (Supplementary Fig. 30). No C-based product was detected in the absence of $H_2O$, indicating the significance of photogenerated holes consumption in the overall $CO_2$ photoreduction. Therefore, both the investigation of $H_2O$ oxidation and $CO_2$ reduction are crucial and directive for the development of photocatalysis. Furthermore, a negligible amount of CO production was detected on $E_7$-$Cu_5Au_1$-$TiO_2$ and $E_7$-$Cu_5Au_1$-$Al_2O_3$ upon 420 nm irradiation (Supplementary Fig. 31a), indicating the inappreciable CuAu LSPR effect and the dominant $TiO_2$ electron donor in photocatalytic $CO_2$ reduction (Supplementary Fig. 31b). Consequently, the same DAs-modified strategy was conducted on the widespread reported photocatalysts of carbon nitride ($C_3N_4$) to further verify the universality of such optimized strategy, and the vectored etching $Cu_5Au_1$ modified $C_3N_4$ and $TiO_2$ also exhibit the efficient $C_2H_4$ production, which sufficiently prove the universality of such CuAu-DAs modification for the optimization of C-C coupling reaction (Supplementary Figs. 32–35).

Cycling tests were performed to analyze the relationship between activity and stability during the photoreduction of $CO_2$ into $C_2H_4$ (Fig. 3d). Based on the limitation of catalyst stability, the majority of photocatalytic $CO_2$ reduction into $C_2H_4$ are maintained for only a few hours and accompany with $C_2H_4$ production stagflation due to the poor structural stability and surface poisoning effect on the photocatalyst (Fig. 3e)[4–13,57]. As shown in Fig. 3e, regardless of the low-efficiency or high-efficiency $C_2H_4$ production rate on different photocatalysts, all exhibited limited reaction time due to the deactivation of photocatalysts. Interestingly, in this work, the yield of each product increases linearly during three cycles of 72 h irradiation, as shown in Fig. 3d, and the AC-HAADF-STEM image of $E_7$-$Cu_5Au_1$-$TiO_2$ displays an unchanged CuAu-DAs structural configuration, as shown in Supplementary Fig. 36, indicating the superior structural stability of $E_7$-$Cu_5Au_1$-$TiO_2$ under such high-efficiency $C_2H_4$ production in the static system. Moreover, compared to the traditional static system, the intermittent flow system is considered a better choice for the improvement of photocatalytic $CO_2$ reduction due to the high-efficiency transport of mass[81–86]. Therefore, the intermittent flow system is also adopted to further analyze the stability of photocatalysts under higher efficiency $CO_2$ conversion circumstances. Notably, in such an intermittent flow system, the $C_2H_4$ production rate reaches $568.8\,\mu mol\cdot g^{-1}\cdot h^{-1}$ (Supplementary Fig. 37a) after 24 h of irradiation, which is superior to those reported in recent works (Fig. 3e). After 5 days of irradiation, $C_2H_4$ production maintains a rate of $483.2\,\mu mol\cdot g^{-1}\cdot h^{-1}$, indicating the superhigh stability of CuAu-DAs catalysts even during such high-efficiency $C_2H_4$ conversion. In addition, a series of long-term photocatalytic stability tests were carried out on different metallic structure-modified $TiO_2$ to determine the reason for the superior stability of such specific CuAu-DAs, and CO temperature-programmed desorption (CO-TPD) was also conducted to further analyze the connection between the catalyst structure and CO affinity. As shown in Fig. 3f, $Cu$-$TiO_2$ shows the strongest CO adsorption strength centered at -120.1 °C in the weak absorption area compared to other catalysts[7], indicating that CO desorption is most difficult on this catalyst surface, which may suppress fresh $CO_2$ adsorption and conversion on $Cu$-$TiO_2$ during the photocatalytic $CO_2$ reduction process. As shown in Supplementary Fig. 38, $Cu$-$TiO_2$ also exhibits nonlinear CO generation, which is ascribed to the partially deactivated Cu sites induced by strong CO absorption. Furthermore, an apparent color change (yellow to black) of $Cu$-$TiO_2$ after 24 h of photocatalytic $CO_2$ reduction is noticed (Supplementary Fig. 38), and the black samples became yellow after 100 °C annealing in a vacuum-treated system (Supplementary Fig. 39), further indicating that the color change could be induced by intermediate adsorption during photocatalytic $CO_2$ reduction. Furthermore, the possible absorbed isotopically labeled $^{13}C$ intermediates of $Cu$-$TiO_2$ after 24 h photocatalytic $CO_2$ reduction and subsequent 100 °C annealing in a vacuum-treated system were analyzed by GC–MS. The GC–MS peak sequences of $^{13}CO$ are shown in Supplementary Fig. 40, and the peak at $m/z = 29$ is assigned to $^{13}CO$, indicating the occurrence of CO poisoning on $Cu$-$TiO_2$, consistent with the results of the CO-TPD analysis and photocatalytic stability test. However, the CO adsorption strength (109.8 °C) of $E_7$-$Cu$-$TiO_2$ is obviously lower than that of $Cu$-$TiO_2$ (Fig. 3f), indicating that the contractible Cu size (nanoparticle to single atom) partially alleviated the CO poisoning effect. Although the CO production of $E_7$-$Cu$-$TiO_2$ increases drastically compared to that of $Cu$-$TiO_2$ due to the decreased Cu coordination number, $E_7$-$Cu$-$TiO_2$ still exhibits nonlinear CO generation and a slight color change (white to pale yellow, Supplementary Fig. 41) due to its slight surface CO poisoning (Supplementary Fig. 42), implying that low-coordination-number Cu inevitably suffers CO poisoning. Notably, the CO adsorption strength of $Cu_5Au_1$-$TiO_2$ is lower than that of $Cu$-$TiO_2$ (Fig. 3f), confirming that the introduction of Au in the Cu lattice weakens the CO adsorption of Cu, as reported[53]. The linearly increased $CH_4$

yield (Supplementary Fig. 43) and weakened CO adsorption (Fig. 3f) of $Cu_5Au_1$-$TiO_2$ compared to Cu-$TiO_2$ suggest that the isolated Au in the Cu lattice efficiently promote *CO conversion to $CH_4$ on the Au sites and suppressed *CO accumulation, which benefits the resistance to catalyst deactivation. However, $Cu_5Au_1$-$TiO_2$ also displays a nonlinearly increased CO yield induced by weak CO poisoning due to the large amount of Cu in the $Cu_5Au_1$ alloy (Supplementary Figs. 43 and 44), which further implies that the addition of Au not only weakens CO adsorption on Cu sites but also serves as a *CO turnover site to rapidly consume the *CO arising from Cu to alleviate the CO poisoning effect on Cu sites. Moreover, the CO-TPD analysis indicates that $E_7$-$Cu_5Au_1$-$TiO_2$ exhibits the highest CO adsorption capacity and lowest CO desorption temperature at -100.0 °C, as shown in Fig. 3f, implying the enormous capacity for CO coverage and the most resistant CO poisoning on CuAu-DAs, which is beneficial for the high efficiency and stability of *CO coupling. Based on the weakest CO chemical adsorption strength of $E_7$-$Cu_5Au_1$-$TiO_2$, no $^{13}$CO of $E_7$-$Cu_5Au_1$-$TiO_2$ was detected by GC–MS after a 24 h stability test and 100 °C annealing in a vacuum-treated system (Supplementary Fig. 45). Meanwhile, there is also no special characteristic peaks of C-based residual surficial absorbates was observed in FTIR spectrum after $CO_2$ photoreduction over $E_7$-$Cu_5Au_1$-$TiO_2$ (Supplementary Fig. 46), suggesting the inexistence of residual C-based intermediate and the superior stability of such CuAu-DAs structure. Owing to the synergistic effect of CuAu-DAs heteronuclear sites, rapid *CO coupling and weakened CO adsorption are simultaneously realized, ensuring superior catalytic sustainability even under such high-efficiency $C_2H_4$ production.

## Mechanism of the photocatalytic performance

Time-dependent in situ diffuse reflectance Fourier transform infrared spectroscopy (DRIFTS) was employed to elucidate the reaction intermediates and concrete evidence of the reaction mechanism under 355 nm laser irradiation for 15 min (Fig. 4a, b). Humid $CO_2$ was carried into the reaction chamber until equilibrium was reached, and different infrared adsorption characteristic peaks of $E_7$-$Cu_5Au_1$-$TiO_2$ were gradually observed when the photocatalyst was subjected to constant irradiation. As shown in Fig. 4b, monodentate carbonate (m-$CO_3^{2-}$ at 1283 cm$^{-1}$) and bidentate carbonate (b-$CO_3^{2-}$ at 1688 cm$^{-1}$) are generated from the co-adsorption of $CO_2$ and $H_2O$ on the surface of $E_7$-$Cu_5Au_1$-$TiO_2$[57,87–90]. The peaks at 1670 and 1646 cm$^{-1}$ are attributed to the vibrations of *$CO_2^-$ and *COOH groups[7,91,92], respectively. Moreover, the characteristic peak at 1660 cm$^{-1}$ is assigned to $H_2O$ decomposition signals[93], and the constantly increased broad IR bands at 3200–3400 cm$^{-1}$ are corresponded to the vibration of *OH groups generated from water dissociation under simulated irradiation[4,6,94]. The different characteristic peaks at approximately 1947 and 2235 cm$^{-1}$ are assigned to *CO intermediates[4,7,95], including Cu-CO and Au-CO on $E_7$-$Cu_5Au_1$-$TiO_2$ (Fig. 4a). Moreover, the contact angles of $Cu_5Au_1$-$TiO_2$ and $E_7$-$Cu_5Au_1$-$TiO_2$ are 9° and 7° (Supplementary Fig. 47), respectively, indicating the better surface hydrophilicity of $E_7$-$Cu_5Au_1$-$TiO_2$, which further demonstrates that the low-coordinated Cu and Au atoms are beneficial for $H_2O$ adsorption and the generation of protons to further facilitate the protonation reaction. Coincidently, asymmetric vibration of *OCCO is observed at -1531 cm$^{-1}$ (Fig. 4b), providing significant evidence for the $C_2$ evolution pathway, indicating that the $C_2$ products arise from the coupling of *CO intermediates[4,94,96,97]. Moreover, the unique C-C coupling intermediates *C=C (3080 cm$^{-1}$)[96], *OCCOH (1579 cm$^{-1}$)[6], *OCCHOH (1307 cm$^{-1}$)[4], and *$C_2H_4$ (1447 cm$^{-1}$)[63] were also detected spectroscopically, further indicating the complexly cascaded multiple electron and proton reaction for the ultimate $C_2H_4$ formation. The vibration frequency of surface-bound *CHO is observed at approximately 1732 and 1710 cm$^{-1}$ on $E_7$-$Cu_5Au_1$-$TiO_2$ due to the one-electron and one-proton reduced reaction of the *CO

intermediate[63,96,98]. Additionally, the peaks at approximately 2993, 2944, and 2881 cm$^{-1}$ are attributed to the C–H symmetric stretching vibrations of methylene (Fig. 4a), facilitating the evolution of $CH_4$ or $C_2H_4$ products[4]. The characteristic spectral peaks at approximately 1372, 1474, and 2965 cm$^{-1}$ are attributed to *$CH_2$, *$CH_2$, and *$CH_3$ intermediates, respectively[57,99], demonstrating that these intermediates could be the source of hydrocarbons. Gibbs free energy theoretical calculations were conducted to elucidate the correlation between the specific nanostructure, electronic properties, and catalytic performance. By combining these results with the time-dependent in situ DRIFTS results, these possible reaction pathways were further proposed by Gibbs free energy calculations. In Fig. 4c, the formations of *$CO_2$, *COOH, and *CO on $E_7$-$Cu_5Au_1$-$TiO_2$ are constantly exothermic and spontaneous processes, indicating that the synergistic effect between Cu-SAs and Au-SAs could significantly promote $CO_2$ adsorption and activation to produce large amounts of *CO intermediates, which is beneficial for *CO coupling under such high concentrations of *CO. The hydrogenation and desorption of *CO to *CHO and CO species require 1.30 and 1.85 eV of energy expenditure, respectively, while *CO coupling to *OCCO consume only 0.54 eV of energy input. Because the energy barrier of *CO coupling is much lower than that of *CO hydrogenation and desorption, the *OCCO intermediates are confirmed to form preferentially during the *CO transformation process. Furthermore, under the constant hydrogenation process of *OCCO, the $C_2H_4$ formation paths are theoretically proposed as described in the following formulas: *OCCO ⟶ *OCCOH ⟶ *OCCHOH ⟶ *OCCH ⟶ *OCCH$_2$ ⟶ *OHCHCH$_2$ ⟶ *CHCH$_2$ ⟶ $C_2H_4$. Therefore, the reaction mechanism of photoreduction $CO_2$ into $C_2H_4$ is determined in detail on the basis of the time-dependent in situ DRIFTS experiments and density functional theory (DFT) simulations.

## Mechanism for the resistance of catalyst deactivation

The enhanced CO poisoning resistance of CuAu hybrid catalysts during photocatalytic $CO_2$ reduction tests compared to that of pure Cu was considered. DFT calculations of Cu-NCs, Cu-SAs, CuAu alloy, and CuAu-DAs modified $TiO_2$ models were performed to further analyze the relation between C-based intermediate conversion efficiency and different nanostructures (Fig. 5a). During the *CO generation process, there is almost no energy input for *$CO_2$, *COOH, and *CO formation in these models (Fig. 5a), which implies rapid *CO formation due to these exothermic and spontaneous processes. Nevertheless, all the conversions from *CO to *CHO, *OCCO, and CO molecules in these models must overcome an enormous energy barrier, indicating the rate-determined significance of *CO conversion during the $CO_2$ reduction process. During three possible *CO conversion routes (*CO ⟶ CO, *CO ⟶ *CHO, *CO ⟶ *OCCO), the optimized energetically favorable *CO coupling modes could not be formed on Cu-NCs-$TiO_2$ (C-C distance, 3.68 Å), Cu-SAs-$TiO_2$ (C-C distance, 2.92 Å), and CuAu-alloy-$TiO_2$ (C-C distance, 3.76 Å) due to the weak interaction between adjacent absorbed *CO intermediates (Fig. 5b). In Supplementary Fig. 48, CuAu-DAs-$TiO_2$ (−1.98 eV) presents the highest *CO adsorption energies compared to CuAu-alloy-$TiO_2$ (−1.44 eV), Cu-NCs-$TiO_2$ (−1.55 eV), and Cu-SAs-$TiO_2$ (−1.79 eV), ensuring the compact *CO interaction and high-concentration *CO coverage[100] (Fig. 5b, C-C distance, 1.40 Å), which contributes to the successful construction of the *OCCO intermediate on CuAu-DAs-$TiO_2$ due to the optimized surface adsorption configurations resulting from the cooperative modification of the steric and electronic properties of CuAu-DAs[26,28–31]. Compared to *CO desorption to CO, *CO is preferentially protonated due to the lower energy barrier and energy input for *CHO production on Cu-NCs-$TiO_2$ (1.15 eV), Cu-SAs-$TiO_2$ (1.41 eV), and CuAu-alloy-$TiO_2$ (0.21 eV). Although the transformation of reaction routes could be an efficient way to resist *CO species accumulation, the energy input for *CO conversion to *CHO on Cu-NCs-$TiO_2$ and Cu-SAs-$TiO_2$ was still

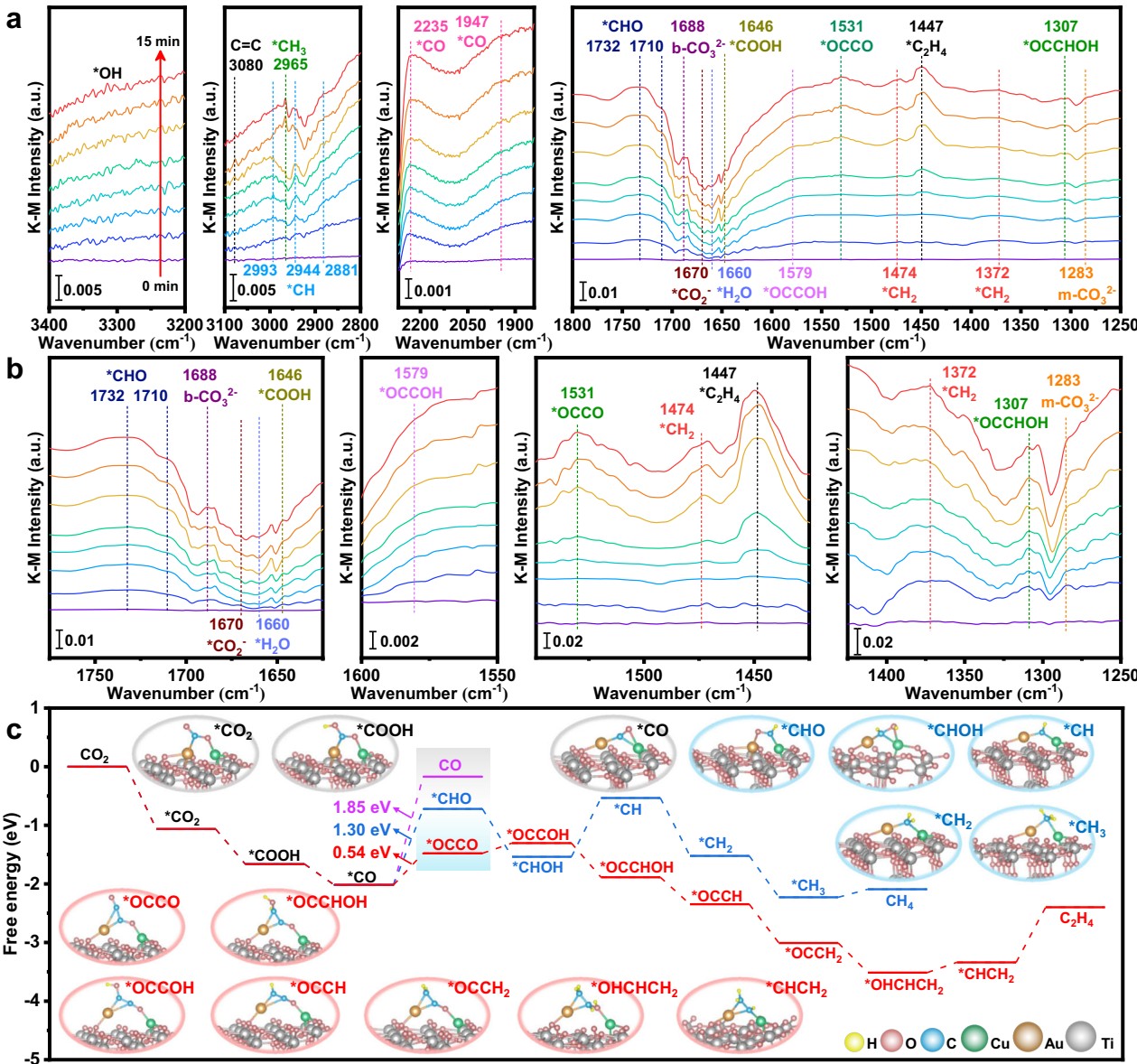

**Fig. 4 | Schematics of $C_2H_4$ generation during photocatalytic $CO_2$ reduction.** **a** In situ DRIFTS detection of $E_7$-$Cu_5Au_1$-$TiO_2$. **b** Corresponding magnifying area of the in situ DRIFTS spectra in **a**. **c** The free energy diagram of $CO_2$ conversion over the CuAu-DAs-$TiO_2$ photocatalyst, together with the atomic structures of the reaction intermediates.

enormous, which also dramatically limited the rapid *CO transformation and led to surface *CO accumulation. Affected by the introduction of Au, the energy input of *CO conversion sharply decreased due to the synergistic cooperation between Cu and Au sites, suggesting that the suppressed *CO poisoning effect could be ascribed to the rapid *CO conversion on CuAu sites[101,102]. Moreover, the energy input for *CO conversion on CuAu-DAs-$TiO_2$ (*CO to *OCCO, 0.54 eV) was much lower than that on Cu-SAs-$TiO_2$ (*CO to *CHO, 1.41 eV) and Cu-NCs-$TiO_2$ (*CO to *CHO, 1.15 eV), which suggested that the extremely low C-C coupling energy barrier-induced rapid *CO consumption could be the key factor for the sharp *CO poisoning resistance on CuAu-DAs-$TiO_2$ (Fig. 5a, c). Owing to the synergistic cooperation between the adjacent Cu-SAs and Au-SAs, simultaneous and high-efficiency *CO formation and *CO conversion could be realized based on the reconstituted surface reactant intermediate adsorption configurations and reduced *OCCO coupling reaction energy barrier, which efficaciously overcomes the activity-stability seesaw effect in photoreduction of $CO_2$ with $H_2O$ into $C_2H_4$ (Fig. 5c).

## Discussion

In summary, CuAu-DAs modified commercial $TiO_2$ was successfully synthesized by an up-bottom atomic synthetic process involving the vectored etching of Cu atoms in a CuAu alloy. According to the activity tests, CO-TPD, and DFT calculation results of the CuAu-DAs structure, Cu-SAs are beneficial for high-efficiency *CO production, while Au-SAs not only moderate the *CO binding strength of the composites but also strikingly increase the *CO coupling efficiency and suppress catalyst deactivation under high-concentration *CO conditions. Owing to such synergistically catalytic effect of heteronuclear DAs, CuAu-DAs-$TiO_2$ exhibited a superhigh rate of 568.8 μmol·$g^{-1}$·$h^{-1}$ in an intermittent flow system, and the negligible CO poisoning of CuAu-DAs-$TiO_2$ was observed during the 120-h photocatalytic stability test. Herein, our discovery not only provides a novel insight into the adjustable synthesis of atomic-level catalysts but also provides a new technique to optimize the selectivity, activity, and stability of photocatalysts based on reconstituted surface adsorption configurations of reactant intermediates and reduced reaction barriers.

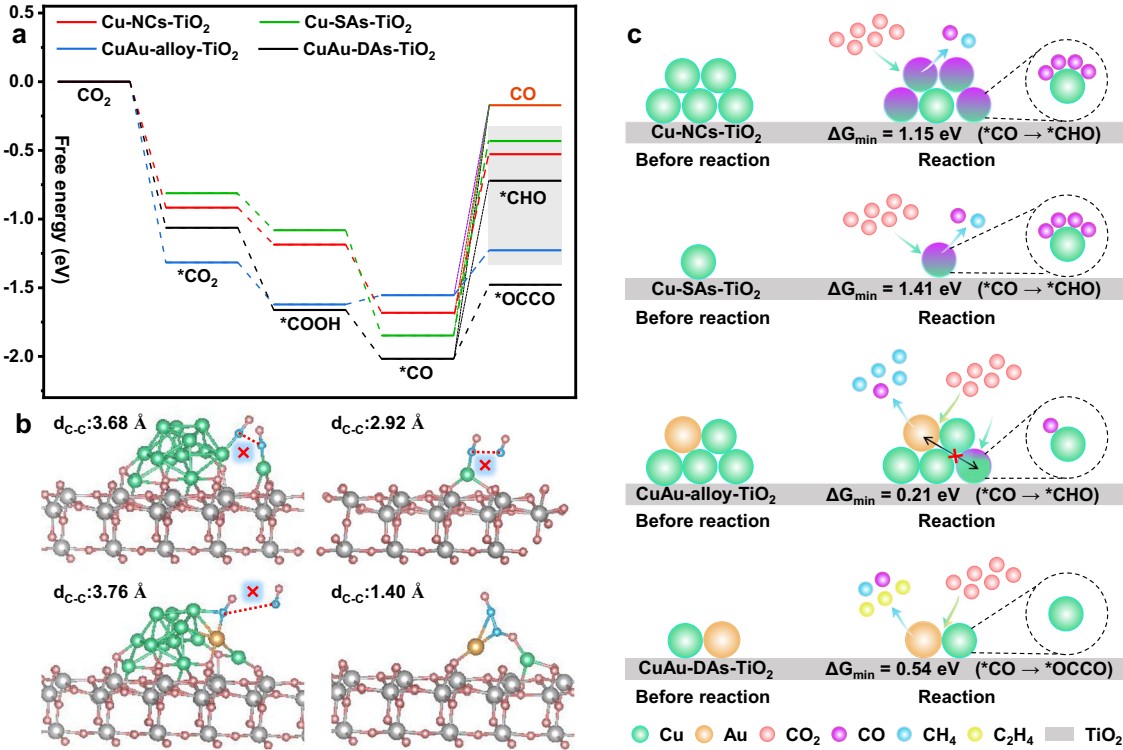

**Fig. 5 | Illustration of the model mechanisms. a** The free energy diagram of $CO_2$ reduction to CO, *CHO, and *OCCO over different modeled surfaces. **b** The most energetically favorable *OCCO adsorption configurations. **c** Schematic illustration of the CO poisoning pathway during the photocatalytic $CO_2$ reduction reaction on different models.

## Methods

### Synthesis of $Cu_5Au_1$-$TiO_2$

Two hundred milligrams of commercial $TiO_2$, 105 mg of PVP, 120 mg of L-ascorbic acid, and 300 mg of KBr were dispersed in 8 mL of deionized water, sonicated for 10 min in a 50 mL round-bottom flask, and transferred into an 80 °C oil bath under constant stirring. After 10 minutes of heating, 0.083 mmol of $CuCl_2 \cdot 2H_2O$ and 0.017 mmol of $HAuCl_4 \cdot 4H_2O$ were dissolved in 4 mL of $H_2O$, and the mixed $CuCl_2 \cdot 2H_2O$ and $HAuCl_4 \cdot 4H_2O$ solution was rapidly transferred into the round-bottom flask. Finally, the solution was kept at 80 °C for 3 h. The sample was washed with deionized water three times and absolute ethanol three times to remove impurities. The obtained samples were denoted as $Cu_5Au_1$-$TiO_2$, which could also be simplified to $C_5A_1$.

### Synthesis of $E_t$-$Cu_5Au_1$-$TiO_2$

$FeCl_3 \cdot 6H_2O$ (0.06 mmol) was dispersed in 10 mL of 0.1 M HCl, and gaseous Ar was bubbled through the solution for 60 min to eliminate dissolved $O_2$. Subsequently, 100 mg of $Cu_5Au_1$-$TiO_2$ was dissolved into the as-prepared solution. The suspension solution was reacted in an oil bath at 50 °C for different etching reaction time (0.5, 1, 3, 5, 7, 9, and 11 h). The as-prepared samples were washed with 0.1 M HCl and deionized water to remove impurities. The obtained samples were denoted as $E_t$-$Cu_5Au_1$-$TiO_2$ (simplified as $E_t$-$C_5A_1$), where t = 0.5, 1, 3, 5, 7, 9 and 11 corresponded to the etching time.

### Synthesis of $E_t$-$Cu_7^N Au_1$-$TiO_2$

$E_t$-$Cu_7^N Au_1$-$TiO_2$ was prepared by the same approach as $E_t$-$Cu_5Au_1$-$TiO_2$, except that the amounts of $CuCl_2 \cdot 2H_2O$ and $FeCl_3 \cdot 6H_2O$ were 0.119 mmol and 0.084 mmol, respectively. The obtained samples were denoted as $E_t$-$Cu_7^N Au_1$-$TiO_2$ (simplified as $E_t$-$C_7^N A_1$), where t = 0.5, 1, 3, 5, 7, 9, 11, and 13 correspond to the etching time.

### Synthesis of $H_t$-$Cu_5Au_1$-$TiO_2$

$H_t$-$Cu_5Au_1$-$TiO_2$ was prepared by the same approach as $E_t$-$Cu_5Au_1$-$TiO_2$ without the addition of $FeCl_3 \cdot 6H_2O$. The obtained samples were denoted as $H_t$-$Cu_5Au_1$-$TiO_2$ (simplified as $H_t$-$C_5A_1$), where t = 0.5, 1, 3, 5, 7, 9, 11, and 13 correspond to the etching time.

### Synthesis of Cu-$TiO_2$

Cu-$TiO_2$ was prepared by the same approach as $Cu_5Au_1$-$TiO_2$ without the addition of $HAuCl_4 \cdot 4H_2O$, and the amount of $CuCl_2 \cdot 2H_2O$ was 0.1 mmol. The obtained sample was denoted as Cu-$TiO_2$ (C-T).

### Synthesis of $E_t$-Cu-$TiO_2$

$E_t$-Cu-$TiO_2$ was prepared by the same approach as $E_t$-$Cu_5Au_1$-$TiO_2$. The obtained samples were denoted as $E_t$-Cu-$TiO_2$ (simplified as $E_t$-C-T), where t = 0.5, 1, 3, 5, 7, 9, and 11 correspond to the etching time.

### Synthesis of Au-$TiO_2$

Au-$TiO_2$ was prepared by the same approach as $Cu_5Au_1$-$TiO_2$ without the addition of $CuCl_2 \cdot 2H_2O$, and the amount of $HAuCl_4 \cdot 4H_2O$ was 0.1 mmol. The obtained sample was denoted as Au-$TiO_2$ (simplified as A-T).

### Synthesis of $E_7$-$Cu_5Au_1$-$Al_2O_3$

$Cu_5Au_1$-$Al_2O_3$ was prepared by the same approach as $E_7$-$Cu_5Au_1$-$TiO_2$. The obtained sample was denoted as $E_7$-$Cu_5Au_1$-$Al_2O_3$.

### Synthesis of $C_3N_4$

$C_3N_4$ was prepared by polymerization method. 10 g of melamine powder was dispersed into 50 mL of deionized aqueous solution and kept in an ultrasonic bath for 30 min. The mixture was stirred in water bath at 50 °C, and then dried in oven at 60 °C. The as-prepared powder was grounded, and then transferred into ceramic crucible calcined at

300 °C for 1 h, 400 °C for 1 h, and 550 °C for 4 h with 2.5 °C·min$^{-1}$ heating rate in a muffle furnace. After natural cooling to room temperature, the samples were washed by DI water for three times to remove impurity.

### Synthesis of Cu$_5$Au$_1$-C$_3$N$_4$

Cu$_5$Au$_1$-C$_3$N$_4$ was prepared by the same approach as Cu$_5$Au$_1$-TiO$_2$, except that the TiO$_2$ was replaced by C$_3$N$_4$.

### Synthesis of E$_t$-Cu$_5$Au$_1$-C$_3$N$_4$

E$_t$-Cu$_5$Au$_1$-C$_3$N$_4$ was prepared by the same approach as E$_t$-Cu$_5$Au$_1$-TiO$_2$, except that the Cu$_5$Au$_1$-TiO$_2$ was replaced by Cu$_5$Au$_1$-C$_3$N$_4$.

### Characterizations

TEM and EDS mapping analysis (Talos F200X G2 200KV) were applied to confirm the morphologies of the materials. AC-HAADF-STEM was conducted on Titan Cubed Themis G2300 and JEM-ARM200F. XAFS of the Cu K-edge and Au L-edge were measured at the BL14W1 station of the Shanghai Synchrotron Radiation Facility (SSRF, China), and Cu foil and Au foil were used as the reference samples. XAFS data were obtained by means of Athena and Artemis software according to standard procedures. ICP–AES was carried out on a Varian VISTA-MPX instrument. The crystal structure was characterized by powder XRD using Cu Kα radiation ($\lambda = 0.15406$ nm) on a Bruker D8 Advance X-ray diffractometer. The compositions of the catalysts were analyzed by XPS (Thermo Scientific K-Alpha). UV–vis–NIR DRS was performed on a Shimadzu UV-2450 spectrophotometer. PL and TRPL analyses were conducted by means of an Edinburgh Instruments instrument (FLS-980). TPD tests were performed on an AutoChem1 II 2920 with a thermal conductivity detector. The surface hydrophilicity of the as-prepared samples was determined by contact angle measurement (CA, KSV CM200, Finland).

### Photoelectrochemical tests

Photocurrent and EIS tests were conducted by a CHI660B electrochemical analyzer with a standard three-electrode system, where Ag/AgCl and Pt wire were used as reference and counter electrodes, respectively, and the samples spin-coated onto fluorine-doped tin oxide (FTO) glasses served as working electrodes. During the photocurrent and EIS tests, 0.2 mol·L$^{-1}$ Na$_2$SO$_4$ solution and 0.1 mol·L$^{-1}$ KCl solution containing 5 mmol·L$^{-1}$ K$_4$[Fe(CN)$_6$]/K$_3$[Fe(CN)$_6$] were applied as electrolytes. Moreover, the Mott-Schottky test was carried out in a 0.5 mol·L$^{-1}$ Na$_2$SO$_4$ solution.

### Photocatalytic activity tests in different reaction systems

(a) Static system: 20 mg of photocatalyst and 5 mL of deionized water were dispersed on a 45 mm diameter quartz glass under sonication and then placed at 60 °C in a vacuum oven for 3 h to dry. Photoreduction CO$_2$ reaction tests were conducted on a closed glass gas-circulation system (Labsolar-6A, Beijing Perfectlight Technology Co., Ltd) with the addition of 2 mL deionized water, and a quartz tray was used to separate water and the quartz glass coated by photocatalysts. The temperature was adjusted to 25 °C using circulating water. (b) intermittent flow system: 2.0 mg of photocatalyst was dispersed on the surface of the microporous membrane with a radius of 2.35 cm by filtration and then sealed in the intermittent flow system with the addition of 0.5 mL deionized water. Before irradiation, the gas-circulation system was vacuum-treated for 15 min and then filled with high-purity CO$_2$ (99.99%). The reactor filled with high-purity CO$_2$ was vacuum-treated again and then filled with high-purity CO$_2$ to reach 90 kPa. The photocatalyst was placed 10 cm away from a 300 W Xe lamp (Microsolar 300, 320–780 nm, 250 mW·cm$^{-2}$, Beijing Perfectlight Technology Co., Ltd). Gas products were detected by means of a gas chromatograph (GC9790II, FULI INSTRUMENTS) equipped with a flame ionization detector (FID) and thermal conductivity detector (TCD). The produced

gases were calibrated with a standard gas mixture, and the identity was determined by the retention time. $^{13}$CO$_2$ isotope labeling experiments were conducted under the same conditions.

### $^{13}$CO$_2$ isotope labeling experiments

The gas products of $^{13}$CO$_2$ isotope labeling experiments were conducted on Agilent 7890B-5977B GC with MS detector. The HP–5MS type column was adopted to analyze the C-based products. The heating programming was started at 40 °C and maintained for 5 min. The temperature of system was increasing from 40 °C to 290 °C with a heating rate of 20 °C·min$^{-1}$, and the system kept 290 °C for 10 min. Helium was used as a carrier gas in GC with a flow rate of 1 mL·min$^{-1}$. The temperature of gas sampling valve is 280 °C under non-diversion mode. Electron impact ionization with 70 eV voltage under the full scan of 2-350 u. The matching degree of CO, CH$_4$, C$_2$H$_4$, and C$_2$H$_6$ relied on the database are about 93, 96, 92, and 93%, respectively, which could be regarded as the reference for source of C-based products.

### In situ DRIFTS experiments

In situ DRIFTS experiments were conducted on a Nicolet iS10 (Thermo) machine, and the photocatalysts were sealed in the reaction chamber with a quartz window. CO$_2$ and H$_2$O were carried into the reaction chamber by N$_2$ flow until equilibrium. After taking the equilibrium system before reaction as the blank background, IR signals were collected under 355 nm laser irradiation (3W) through the quartz glass window.

### XAFS date analysis

Data reduction, data analysis, and EXAFS fitting were performed and analyzed with the Athena and Artemis programs of the Demeter data analysis packages[103] that utilizes the FEFF6 program[104] to fit the EXAFS data. The energy calibration of the sample was conducted through a standard Cu foil, which as a reference was simultaneously measured. A linear function was subtracted from the pre-edge region, then the edge jump was normalized using Athena software. The k$^3$-weighted χ(k) data were Fourier transformed after applying a Kaiser-Bessel window function ($\Delta k = 1.0$). For EXAFS modeling, the global amplitude EXAFS (CN, R, $\sigma^2$ and $\Delta E_0$) were obtained by nonlinear fitting, with least-squares refinement, of the EXAFS equation to the Fourier-transformed data in R-space, using Artemis software, EXAFS of the Cu foil is fitted and the obtained amplitude reduction factor $S_0^2$ value (0.845) was set in the EXAFS analysis to determine the coordination numbers (CNs) in the Cu-O and Cu-Au scattering path in sample.

### Computational methods

DFT models were constructed based on the results of AC-HAADF-STEM and XRD, and the (101) crystal plane of TiO$_2$ was selected to construct the computational models. All DFT calculations were performed by the Vienna Ab Initio Simulation Package (VASP). The Perdew-Burke-Ernzerhof (PBE) generalized gradient approximation (GGA) was utilized to address the exchange-correlation interactions. All slab models were applied with a 15 Å vacuum layer to prevent interactions between slabs. A $2 \times 2 \times 1$ k-point was sampled in the Brillouin zone. A plane-wave basis expansion with a 400 eV energy cutoff was adopted to optimize the geometric structures, and the electronic forces were converged to $1 \times 10^{-5}$ and 0.03 eV·Å$^{-1}$, respectively. The Gibbs free energies were calculated at 298.15 K by means of the formula $G = E_{DFT} - TS + E_{ZPE}$, where $E_{DFT}$, $TS$, and $E_{ZPE}$ represent the electronic energy of each step, entropy contribution, and zero-point energy, respectively.

## Data availability

All data that support the findings of this study are present in the paper and the Supplementary Information. Further information can be

acquired from the corresponding authors. Source data are provided with this paper.

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

## Acknowledgements

This work was supported by the National Natural Science Foundation of China (22225808), Sino-German Cooperation Group Project (GZ1579), National Natural Science Foundation of China (22025506), National Natural Science Foundation of China (22075111), Jiangsu Province Innovation Support Program International Science and Technology Cooperation Project (BZ2022045), the Industry Prospect and Key Core Technology (Competition Project) of Jiangsu Province (BE2019093), and National Science and Technology Major Project of China (2021YFB3500700).

## Author contributions

Z.X. and W.S. conceived the project and wrote the manuscript. S.S. pointed and revised the shortage during this work. S.X. and X.Wu designed the synthesis of composite samples. L.L. performed the computational calculation. S.G. dealt with the XANES analysis. D.X., B.M., and T.Z. helped with the data collection and analysis. M.C. and X.Wang oversaw the project. All authors discussed and commented on the manuscript.

## Competing interests

The authors declare no competing interests.
