## [Peer Review File · Nature Communications]

Well-defined diatomic catalysis for photosynthesis of C₂H₄ from CO₂Reviewers' comments:

Reviewer #1 (Remarks to the Author):

Recommendation: Publish elsewhere.

Comments:

In this manuscript, the authors have reported the construction of the TiO₂ based CuAu-DACs for photochemical conversion of CO₂ to C₂H₄. Overall, the authors have well characterized the materials, evaluated the photocatalytic performance, and explored the possible reaction mechanism. However, some key issues are still needs to be addressed in this work, and therefore, I cannot support the publication of this work in Nature Communications. The specific comments are provided for the authors to consider for improving this work.

1. What about the water oxidation half reaction? I am confused that did the authors determine the generated O₂ in the half reaction? O₂ should be determined carefully and exactly to check the stoichiometry of the photogenerated electrons and holes.
2. The auctors have used 20 mg of catalyst during photocatalytic CO₂ reduction reaction. Are there any changes in catalytic activity with altering the catalyst amount, due to the limitation of mass transfer? Author should carefully mention this in the article.
3. It appears H₂O is also critically involved in the reaction mechanism. Please clarify the role of water in the overall reaction mechanism presented in Figure 4.
4. How do understand the longer PL lifetime of the CuAu-DACs than the other catalysts indicate the recombination of photoinduced charges is suppressed in the DACs? Many previous works highlighted that photocatalysts show shortened PL lifetime compared to individuals (J. Am. Chem. Soc. 2017, 139, 16, 5660; J. Am. Chem. Soc. 2018, 140, 48, 16514; Nat. Commun. 2020, 11, 1149).
5. It is recommended to use g-C₃N₄ as a support for the synthesis of CuAu-DACs photocatalyst and compare the activity with TiO₂ supported catalyst. Is

there any particular reason behind choosing the TiO₂ support as catalyst.

6. The details and parameters of the GC-MS analysis should be provided, especially the method/mode for resolving the total ion flow and the corresponding matching degree.

7. In Figure 4a, in-situ DRIFTS spectra I didn't find any clear peak for *OCCOH, *CH₂, *OCCHOH and m-CO₃²⁻ intermediates. The authors must have zoomed in these regions and separately show the peaks. Similar thing I have noticed for *CO also. It is highly recommended to revisit the DRIFTS spectra to get more refined spectrum. For reference, please cite this recently published article <https://doi.org/10.1002/anie.202311304>.

Reviewer #2 (Remarks to the Author):

In the manuscript of "Well-defined diatomic catalysis for photosynthesis of C₂H₄ from CO₂", Xie et. al. reported a up-bottom synthetic strategy for the fabrication of ordered CuAu diatomic photocatalysts, and this synthetic approach shows great potential for the construction of high-efficiency atomic level catalysts. The robust catalyst CuAu-DAs-TiO₂ exhibited efficient performance for photocatalytic CO₂ conversion to C₂H₄, and mechanism research demonstrate that synergistic effects between Au and Cu single atom sites contributed this enhanced photocatalytic activity. Therefore, I would like to recommend for publication of this work in the journal of Nature Communications after addressing the following issues:

(1) Why such up-bottom synthetic approach could regularly construct the heteronuclear diatomics? Is there interaction between different metal atoms, which results in the ordered formation of the diatomic pair. Is there any change of the atomic distance between Au and Cu single atoms before and after the vectored etching process?

(2) In Figure 3d, with the reaction time prolonging, CH₄ production increased deeply compared to other products, the authors should give a reasonable explanation for this phenomenon.

(3) In Supplementary Figure 18, all the H-Cu₅Au₁-TiO₂ display the enhanced C₂H₆ production, while the negligible C₂H₄ increase was also observed, which is apparently different from those E-Cu₅Au₁-TiO₂ photocatalysts. It is suggested to clarify such difference from structural configuration.

(4) There are many C-based intermediates in the whole CO₂ photoreduction process. Why the author considers the *CO is the predominate adsorbate, please give more evidence.

(5) It is suggested to perform the FTIR spectrum of E7-CuAu-TiO₂ before and after CO₂ photoreduction to analyze the possible C-based residual surficial adsorbates.

(6) The methods of the XAFS data fitting should be added in the manuscript, which is significant reference.

(7) From the photocatalytic results, it can be obviously observed that the C₂H₄ is the main product for E1-C₅A₁, however, CH₄ is the main product for C₅A₁ and H11-C₅A₁. The authors must provide more experimental characterization and explanations to clarify this point. This will provide readers more useful information to design efficient and selective catalysts

Reviewer #3 (Remarks to the Author):

In this manuscript, the authors claim to have synthesized Cu-Au dual atoms (DAs) on TiO₂ via a selective leaching approach and investigate their efficacy in photoconversion of CO₂. They conducted extensive studies but the results seem not to sufficiently support the conclusions and thoroughly explored the beneficial effects of Cu-Au DAs in the photocatalytic reduction of CO₂. Although the performance seems very promising the formation mechanism and identification of Cu-Au DAs is not convincing. Therefore, I don't think the manuscript can be accepted in Nature Communications.

1. Necessary characterization of commercial TiO₂ such as SEM etc. should be added for better comparison with Cu₅Au₁-TiO₂.

2. The identification of Cu SAs and Au SAs from the STEM images is not

convincing. The resolution of the images is too low to identify the SAs (e.g. Figure 1c,d), let alone to distinguish Au and Cu from the negligible difference in contrast and size.

3. In the E11-Cu₅Au₁-TiO₂ sample, it is obvious from the ICP result that Cu is still present, therefore in Figure 1g CuSAs may also be present and Figure 1a is not correct.

4. In a typical leaching process, one might anticipate the initial leaching of Cu on the outer surface of Cu₅Au₁. Subsequently, it's expected that the surrounding Au atoms bonded with those Cu atoms would also be affected. However, based on the ICP results, the amount of Au appears to remain constant. This observation requires a clear and detailed demonstration.

5. XRD analysis seems inconclusive. i. In ref 62, the XRD peak of AuCu is close to Au because Au is the main component, but in this study, Cu is the main component. ii. The observation of Au peaks is an obvious sign for the existence of Au nanoparticles. iii. There are no Cu diffraction peaks in the XRD pattern of Cu-TiO₂ sample, and the authors attribute this to the ultrafine distribution of Cu, but there is no characterization to support this.

6. The photocurrent spectra in Figure S12 look strange. The authors should provide some explanations.

7. Using UV-Vis spectra to determine the bandgap is not precise. And it is also not reasonable to determine the band gap of TiO₂ deposited with metals.

8. In XANES spectra, the Au L-edge shows only similar spectra to Au₀ in contrast to the Cu K-edge in which the spectra is different from Cu₀. Also, why is the Cu-Au peak in the Cu K-edge relatively smaller than the Au-Cu peak which is predominant in the Au L-edge?

9. In PL spectroscopy, Cu₅Au₁-TiO₂ shows similar lifetime to Cu-TiO₂. It is not clear that Au addition improves charge transfer efficiency.

Dear reviewers,

We are appreciative for the reviewers' insightful and constructive feedback. We recognize that the reviewers' guidance has enabled us to produce a greatly improved manuscript, whose novelty and advance are now more readily convincing in the introduction and discussion. Based on the reviewers' comments and concerns, we have made extensive alterations to the structure, mechanism, presentation, and analysis of our findings. Herein, we hope our work could be evaluated again in *Nature Communications* based on the point-to-point resolution of all the reviewers' concerns.

Reviewers#1' comments:

In this manuscript, the authors have reported the construction of the TiO₂ based CuAu-DACs for photochemical conversion of CO₂ to C₂H₄. Overall, the authors have well characterized the materials, evaluated the photocatalytic performance, and explored the possible reaction mechanism. However, some key issues are still needs to be addressed in this work, and therefore, I cannot support the publication of this work in Nature Communications. The specific comments are provided for the authors to consider for improving this work.

Response to comments:

Thank you for your approbation about our work, especially in the aspects of material characterization, photocatalytic performance, reaction mechanism. Moreover, these comments and suggestions have not only enabled us to provide a highly improved manuscript but also inspired us to conduct more in-depth studies. Finally, the dominated concerns related to the mechanism of H₂O oxidation and the universality of diatomic strategy have been addressed by the experimental replenishment. Therefore, after addressing all these issues, we hope the work focused on such specific design of the well-defined DAs with high-efficiency catalytic performance could be further reconsidered to publish in Nature Communications.

Q1. What about the water oxidation half reaction? I am confused that did the authors determine the generated O₂ in the half reaction? O₂ should be determined carefully and

exactly to check the stoichiometry of the photogenerated electrons and holes.

Response:

Thank you for your careful revision and point the shortage. Overall CO₂ photoreduction is divided into two major half reaction steps, the CO₂ reduction and H₂O oxidation [1,2]. Therefore, the H₂O oxidation process is also significant for deep-going understanding of overall CO₂ photoreduction due to the matched relationship between photogenerated electrons and holes. Here, we qualitate and quantify the O₂ evolution by TCD detector in gas chromatograph (GC9790II, FULI INSTRUMENTS) under the calibration of standard pure O₂. As shown in **Figure R1**, O₂ evolution related to photogenerated holes consumption is stoichiometrically approximate to products of photogenerated electrons reduction (holes : electrons \approx 1 : 1). Moreover, H₂¹⁶O was replaced by H₂¹⁸O to trace the source of O₂ during the CO₂ photoreduction process by GC–MS. The detection of ¹⁶O¹⁸O and ¹⁸O₂ species verifies that O₂ originates from H₂O oxidation in the photocatalytic CO₂ reduction. The corresponding revision and discussion have been added in manuscript (line 378-383, Supplementary Figs. 28 and 29, References 12 and 80).

Figure R1. O₂ production rates of different TiO₂-based samples.

Figure R2. GC-MS spectra of CO, CH₄, C₂H₄, and C₂H₆ from the photocatalytic reduction of the $^{13}\text{CO}_2$ and $\text{H}_2\text{}^{18}\text{O}$ isotope labeling experiments on E₇-Cu₅Au₁-TiO₂.

[1] Das, R. et al. Engineering the charge density on an In_{2.77}S₄/porous organic polymer hybrid photocatalyst for CO₂-to-ethylene conversion reaction. *J. Am. Chem. Soc.* **145**, 422–435 (2023).

[2] Wang, Y. J. & He, T. Recent advances and comprehensive consideration on the oxidation half reaction in photocatalytic CO₂ conversion. *J. Mater. Chem. A* **9**, 87–110 (2021).

Q2. The authors have used 20 mg of catalyst during photocatalytic CO₂ reduction reaction. Are there any changes in catalytic activity with altering the catalyst amount, due to the limitation of mass transfer? Author should carefully mention this in the article.

Response:

Thank you for your careful consideration. The optimized amount of photocatalysts was verified to be 20 mg with a highest C₂H₄ production rate based on the excellent mass transfer and superior light utilization, and the apparent decrease of C-based products production rate was observed with the addition of superfluous photocatalysts, which could be ascribed to the impeded light transmission [1]. The corresponding revision and discussion have been added in manuscript (line 337-342, Supplementary Fig. 23, References 74).

Figure R3. Dependence of the photocatalytic CO₂ reduction rate over different amounts of E₇-Cu₅Au₁-TiO₂.

[1] Wang, E. R. Y. et al. Solar-driven photocatalytic reforming of lignocellulose into H₂ and value-added biochemicals. *ACS Catal.* **12**, 11206–11215 (2022).

Q3. It appears H₂O is also critically involved in the reaction mechanism. Please clarify the role of water in the overall reaction mechanism presented in Figure 4.

Response:

Thank you very much for your rigorous consideration. In a typical overall CO₂ photoreduction process, H₂O molecules not only serve as proton donors, but also respond to consume photogenerated holes [1-3]. Usually, H₂O is accustomed to decomposing into proton and hydroxyl, which are responsible for protonation of C-based intermediates and elimination of photogenerated holes, respectively [4]. As shown in **Figure R4**, lots of hydrocarbon intermediates were detected in the *in situ* DRIFTS spectrum, such as *COOH, *CHO, *CH, *CH₂, *CH₃, *OCCOH, *OCCHOH, and *C₂H₄, suggesting the significant protonation steps during the whole CO₂ reduction process. Moreover, the characteristic peak at 1660 cm⁻¹ is assigned to H₂O decomposition signals [4], and the constantly increased broad IR bands at 3200-3400

cm^{-1} are corresponded to the vibration of $^*\text{OH}$ groups generated from water dissociation under simulated irradiation [5,6]. The CO_2 photoreduction experiment was also taken under no existence of H_2O to further figure out the influence of H_2O species (**Figure R5**). No C-based product was detected in the absence of H_2O , indicating the significance of photogenerated holes consumption in the overall CO_2 photoreduction. Therefore, both the investigation of H_2O oxidation and CO_2 reduction are crucial and directive for the development of photocatalysis. The corresponding revision and discussion have been added in manuscript (line 383-388 and 487-490, Supplementary Figs. 29 and 30, References 4, 6, and 94).

Figure R4. a, *In situ* DRIFTS detection of $\text{E}_7\text{-Cu}_5\text{Au}_1\text{-TiO}_2$. **b**, Corresponding magnifying area in *in situ* DRIFTS spectra of (a).

Figure R5. Comparison of CO₂ photoreduction with or without H₂O addition.

[1] Das, R., Chakraborty, S. & Peter, S. C. Systematic assessment of solvent selection in photocatalytic CO₂ reduction. *ACS Energy Lett.* **6**, 3270–3274 (2021).

[2] Das, R. et al. Engineering the charge density on an In_{2.77}S₄/porous organic polymer hybrid photocatalyst for CO₂-to-ethylene conversion reaction. *J. Am. Chem. Soc.* **145**, 422–435 (2023).

[3] Nosaka, Y. & Nosaka, A. Y. Generation and detection of reactive oxygen species in photocatalysis. *Chem Rev.* **117**, 11302–11336 (2017).

[4] Yin, S. K. et al. Boosting water decomposition by sulfur vacancies for efficient CO₂ photoreduction. *Energy Environ. Sci.* **15**, 1556–1562 (2022).

[5] Gao, W. et al. Vacancy-defect modulated pathway of photoreduction of CO₂ on single atomically thin AgInP₂S₆ sheets into olefiant gas. *Nat. Commun.* **12**, 4747 (2021).

[6] Wang, W. et al. Photocatalytic C–C coupling from carbon dioxide reduction on copper oxide with mixed-valence copper(I)/copper(II). *J. Am. Chem. Soc.* **143**, 2984–2993 (2021).

Q4. How do understand the longer PL lifetime of the CuAu-DACs than the other catalysts indicate the recombination of photoinduced charges is suppressed in the DACs? Many previous works highlighted that photocatalysts show shortened PL lifetime compared to individuals (*J. Am. Chem. Soc.* 2017, 139, 16, 5660; *J. Am. Chem. Soc.* 2018, 140, 48, 16514; *Nat. Commun.* 2020, 11, 1149).

Response:

Thank you for your considerate advice. In detail, the result of our work is matched well with the referred works proposed by reviewer, and we also replenish the more detailed description according to the references that provided by reviewer. It is reported that heterogeneous photocatalysts possess the faster PL decay and shorter PL lifetime than individuals due to the sluggish surface charge recombination [1-3]. The contributions of τ_1 and τ_2 are more related to charge transfer, and the PL decay is more dominated by τ_3 [1-3]. The CuAu-DAs modified TiO₂ exhibits shorter τ_1 , τ_2 , and τ_3 compared to TiO₂, illustrating the compact interaction and suppressed charge recombination between CuAu-DAs and TiO₂. Moreover, the shortest average PL decay time of CuAu-DAs modified TiO₂ represents the fastest decay in CuAu-DAs-TiO₂, which is ascribed to the fact that the direct connection between CuAu-DAs and TiO₂ is more beneficial for convenient photogenerated charge transfer from TiO₂ to CuAu-DAs rather than the recombination in Cu and Cu₅Au₁ bulk. The corresponding revision and discussion have been added in manuscript (line 274-282, References 67-69).

Photocatalysts	A ₁	τ_1 /ns	A ₂	τ_2 /ns	A ₃	T ₃ /ns	τ /ns
TiO ₂	730.23	1.23	209.76	6.24	168.95	47.39	38.05
Cu-TiO ₂	1193.29	0.60	45.55	3.87	18.30	39.42	18.25
Cu ₅ Au ₁ -TiO ₂	1038.55	1.05	115.17	4.44	39.85	35.69	17.94
E ₇ -Cu ₅ Au ₁ -TiO ₂	1097.26	0.79	104.96	3.18	10.07	29.44	6.99

Table R1. The time-resolved fluorescence spectra of as-prepared samples.

[1] Xu, Y. et al. A CsPbBr₃ perovskite quantum dot/graphene oxide composite for photocatalytic CO₂ reduction. *J. Am. Chem. Soc.* **139**, 5660–5663 (2017).

[2] Cui, X. et al. Turning Au nanoclusters catalytically active for visible-light driven CO₂ reduction through bridging ligands. *J. Am. Chem. Soc.* **140**, 16514–16520 (2018).

[3] Wang, S. et al. Intermolecular cascaded π -conjugation channels for electron delivery powering CO₂ photoreduction. *Nat. Commun.* **11**, 1149 (2020).

[4] Lightcap, I. V. & Kamat, P. V. Fortification of CdSe quantum dots with graphene oxide. excited state interactions and light energy conversion. *J. Am. Chem. Soc.* **134**, 7109–7116 (2012).

Q5. It is recommended to use g-C₃N₄ as a support for the synthesis of CuAu-DACs photocatalyst and compare the activity with TiO₂ supported catalyst. Is there any particular reason behind choosing the TiO₂ support as catalyst.

Response:

Thank you for your expansive suggestion. Commercial TiO₂ has been reported as the suitable redox capacity, non-poisonous, and low-cost industrialized products, indicating that the development of efficient TiO₂-based photocatalysts could be great potential for the photocatalytic industrialized process. Moreover, TiO₂ also serves as a typical metallic oxide photocatalyst, the successful optimized strategies on TiO₂ could be universality on most other metallic oxide photocatalysts. Based on the unique electronic properties, facile synthesis, excellent chemical durability, and adjustable bandgap of metal-free polymeric carbon nitride (C₃N₄), C₃N₄ also has been regarded as widespread reported typical photocatalysts [1,2]. Moreover, the same DAs-modified strategy was conducted on C₃N₄ to further verify the universality of such optimized strategy. As shown in **Figure R6a-c**, the CuAu alloy nanoparticle was uniformly dispersed on the surface of C₃N₄ nanosheet (**Figure R6a,b**), and the lattice spacing of 0.202, 0.231, 0.238, and 0.236 are ascribed to (200), (111), (111), and (111) plane of Au (**Figure R6c**). However, HAADF-STEM (**Figure R6d**) and the corresponding elemental mapping (**Figure R5e-h**) verify the superimposed Cu and Au elements distribution, indicating the existence of CuAu alloy. Despite the EDS analysis (**Figure R7**) displays the 4.42 of Cu/Au atomic ratio, there is still no Cu lattice exhibited due to the nondominated Miller indexes of Cu phase (**Figure R8a**). The characteristic peaks at ca. 12.8 (100) and 27.5° (002) are attributed to the similar parallel diffraction modes and in-plane structure with pristine C₃N₄ [1,2], indicating the successful fabrication of C₃N₄. Notably, the intensity of Au XRD characteristic peaks and Au LSPR response first increase and then decrease, which implies that more Au sites were exposed due to the vectored etching of Cu elements, while the Au lattice was final to disappear and distribute as Au single atoms based on the constant decomposition of Cu lattice. To further verify the universality of such optimized strategy for CO₂ photoreduction into C₂H₄, the CO₂

photoreduction tests were also conducted on C_3N_4 -based samples under 320-780 nm irradiation (**Figure R9a,b**). After 9 h vectored etching of Cu in $Cu_5Au_1-C_3N_4$, the $E_9-Cu_5Au_1-C_3N_4$ exhibits the highest C_2H_4 production at a rate of $17.8 \mu\text{mol}\cdot\text{g}^{-1}\cdot\text{h}^{-1}$, and the selectivity of C_2H_4 reaches up to 68%, which also suggests that the low-coordination adjacent Cu and Au atoms could benefit for the C-C coupling and C_2H_4 production. Although the $E_9-Cu_5Au_1-C_3N_4$ exhibits the highly selective C_2H_4 production, the C_2H_4 evolution rate is still lower than that of TiO_2 -based photocatalysts due to the natural difference between C_3N_4 and TiO_2 , and the specific reasons for such difference will be explored in our future work. Consequently, both the vectored etching Cu_5Au_1 modified C_3N_4 and TiO_2 exhibit the high C_2H_4 selectivity, which sufficiently prove the universality of such CuAu heteronuclear diatomic modification for the optimization of C-C coupling reaction. The corresponding revision and discussion have been added in manuscript (line 392-397 and 628-637, Supplementary Figs. 32-35).

Figure R6. **a**, TEM image of C_3N_4 . **b**, TEM image of $Cu_5Au_1-C_3N_4$. **c**, HRTEM image of $Cu_5Au_1-C_3N_4$ that magnified in (b). **d**, HAADF-STEM image of $Cu_5Au_1-C_3N_4$. **e-h**, The corresponding EDS mapping images of $Cu_5Au_1-C_3N_4$ in (d).

Figure R7. a, HAADF-STEM image of $\text{Cu}_5\text{Au}_1\text{-C}_3\text{N}_4$. b, EDS spectrum of the selected area in (a).

Figure R8. a, XRD patterns of $\text{Cu}_5\text{Au}_1\text{-C}_3\text{N}_4$ with different vectored etching time. b, UV-vis-NIR DRS spectra of $\text{Cu}_5\text{Au}_1\text{-C}_3\text{N}_4$ with different vectored etching time.

Figure R9. a, CH_4 , C_2H_4 , C_2H_6 , CO, and H_2 production rates of $\text{E}_T\text{-Cu}_5\text{Au}_1\text{-C}_3\text{N}_4$. b, Electron-based selectivity of photocatalytic CO_2 conversion over $\text{E}_T\text{-Cu}_5\text{Au}_1\text{-C}_3\text{N}_4$.

[1] Xie, Z. K. et al. Near-infrared light-driven photocatalytic reforming lignocellulose into H_2 and chemicals over heterogeneous carbon nitride. *ACS Catal.* **13**, 13768–13776 (2023).

[2] Chen, Z. H. et al. Efficient light-free activation of peroxymonosulfate by carbon

ring conjugated carbon nitride for elimination of organic pollutants. *Chem. Eng. J.* **420**, 129671 (2021).

Q6. The details and parameters of the GC-MS analysis should be provided, especially the method/mode for resolving the total ion flow and the corresponding matching degree.

Response:

Thank you for your scrupulous examination and good propose. The $^{13}\text{CO}_2$ isotope labeling experiments were conducted on Agilent 7890B-5977B GC with MS detector. The HP-5MS type column was adopted to analyze the C-based products. The heating programming was started at 40 °C and maintained for 5 min. The temperature of system was increasing from 40 °C to 290 °C with a heating rate of 20 °C/min, and the system kept 290 °C for 10 min. Helium was used as a carrier gas in GC with a flow rate of 1 mL/min. The temperature of gas sampling valve is 280 °C under non-diversion mode. Electron impact ionization with 70 eV voltage under the full scan of 2-350 u. The matching degree of CO, CH₄, C₂H₄, and C₂H₆ relied on the database are about 93, 96, 92, and 93%, respectively, which could be regarded as the reference for source of C-based products. The corresponding revision and discussion have been added in manuscript (line 675-683).

Q7. In Figure 4a, in-situ DRIFTS spectra I didn't find any clear peak for *OCCOH, *CH₂, *OCCHOH and m-CO₃²⁻ intermediates. The authors must have zoomed in these regions and separately show the peaks. Similar thing I have noticed for *CO also. It is highly recommended to revisit the DRIFTS spectra to get more refined spectrum. For reference, please cite this recently published article <https://doi.org/10.1002/anie.202311304>.

Response:

Thank you for your excellent suggestion for the optimization of our work. After the studying about the mentioned work, we found these works (*Angew. Chem. Int. Ed.* 2023, e202311304; *Small* 2023, 2305307) are significant for the guidance of our work,

especially for the part of DRIFTS [1,2]. Here, we have cited these works and revisit the section of DRIFTS in manuscript to further optimize our research (**Figure R10**). The corresponding revision and discussion have been added in manuscript (line 484-487, References 89 and 90).

[1] Paul, R. et al. Tweaking photo CO₂ reduction by altering lewis acidic sites in metalated-porous organic polymer for adjustable H₂/CO ratio in syngas production. *Angew. Chem. Int. Ed.* e202311304 (2023).

[2] Paul, R. et al. Pyrolysis free out-of-plane Co-single atomic sites in porous organic photopolymer stimulates solar-powered CO₂ fixation. *Small*, 2305307 (2023).

Figure R10. a, *In situ* DRIFTS detection of E₇-Cu₅Au₁-TiO₂. b, Corresponding magnifying area in *in situ* DRIFTS spectra of (a).

Reviewer #2 (Remarks to the Author):

In the manuscript of “Well-defined diatomic catalysis for photosynthesis of C₂H₄ from CO₂”, Xie et. al. reported an up-bottom synthetic strategy for the fabrication of ordered CuAu diatomic photocatalysts, and this synthetic approach shows great potential for the construction of high-efficiency atomic level catalysts. The robust catalyst CuAu-DAs-TiO₂ exhibited efficient performance for photocatalytic CO₂ conversion to C₂H₄,

and mechanism research demonstrate that synergistic effects between Au and Cu single atom sites contributed this enhanced photocatalytic activity. Therefore, I would like to recommend for publication of this work in the journal of Nature Communications after addressing the following issues:

Response to comments:

Thank you for the commend about the advance and interest in our work. We have adopted the suggestion and revised all the issues that you proposed. We hope the replenished dates and explanation could reach up to your request.

Q1. Why such up-bottom synthetic approach could regularly construct the heteronuclear diatomics? Is there interaction between different metal atoms, which results in the ordered formation of the diatomic pair. Is there any change of the atomic distance between Au and Cu single atoms before and after the vectored etching process?

Response:

Thank you very much for your rigorous consideration. Sikorska et al. reported the intermolecular potential related interparticle distance [1], and two particles attract each other at the range of special distance [1-3]. Here, the compact interaction could be produced from the heteronuclear metal atoms, which contributes to the tangle between Au and Cu atoms. Moreover, with the decomposition of CuAu alloy, the distance between Cu and Au increased a little (**Figure R1,2**), indicating the weakened interaction between Cu and Au atoms due to the CuAu alloy dissolution. The corresponding revision and discussion have been added in manuscript (**line 100-106, Supplementary Fig. 4a-c**).

Figure R1. Acquired AC-HAADF-STEM images intensity profile of (a) $\text{Cu}_5\text{Au}_1\text{-TiO}_2$ for the analysis of (b) Au-Au distance and (c) Au-Cu distance.

Figure R2. Acquired AC-HAADF-STEM images intensity profile of $\text{E}_7\text{-Cu}_5\text{Au}_1\text{-TiO}_2$ for the analysis of Cu-Au distance.

[1] Sikorska, C. & Gaston, N. Modified Lennard-Jones potentials for nanoscale atoms. *J. Comput. Chem.* **41**, 1985-2000 (2020).

[2] Rose, J. et al. Universal features of the equation of state of metals. *Phys. Rev. B* **29**, 2963 (1984).

[3] Erkoç, S. Empirical many-body potential energy functions used in computer

simulations of condensed matter properties. *Phys. Rep.* **278**, 79-105 (1997).

Q2. In Figure 3d, with the reaction time prolonging, CH₄ production increased deeply compared to other products, the authors should give a reasonable explanation for this phenomenon.

Response:

Thank you for your considerate guidance. Photocatalytic CO₂ reduction selectivity is often directly depended on the structure of active sites. In this work, H_T-CuAu-TiO₂ displays superior CH₄ production and selectivity, indicating such CuAu clustering structure benefits for CH₄ evolution. Moreover, there has been reported that light-induced agglomeration of single-atom platinum in photocatalyst [1]. The light induced destabilization of the single-atom Pt species to binding of surface-coordinated Pt with solution-hydrogen (adsorbed H atoms), which consequently weakens the Pt single-atom bonding to the TiO₂ surface. Therefore, we speculate such CuAu-DAs structure could also be changeable during the reaction process due to the complex intermediates during the CO₂ photoreduction, which contributes to the change of photocatalytic selectivity.

[1] Denisov, N. et al. Light-induced agglomeration of single-atom platinum in photocatalysis. *Adv. Mater.* **35**, 2206569 (2023).

Q3. In Supplementary Figure 18, all the H-Cu₅Au₁-TiO₂ display the enhanced C₂H₆ production, while the negligible C₂H₄ increase was also observed, which is apparently different from those E-Cu₅Au₁-TiO₂ photocatalysts. It is suggested to clarify such difference from structural configuration.

Response:

Thank you for your careful check and good suggestion. The selectivity of CO₂ reduction products is mainly ascribed to the changed structure of CuAu. Usually, most researches mainly concentrate on manipulating the Cu structure for CO₂ absorption and activation, with limited studies on the influence of rapid surface hydrogen radicals (*H) production through the swift H₂O dissociation and proton (H⁺) reduction [1-5]. Therefore, compared to E₇-Cu₅Au₁-TiO₂, H₁₁-Cu₅Au₁-TiO₂ possesses much more Cu active sites

(Figure R3), benefiting for the more *H production. Owing to the abundant surficial *H of $H_{11}-Cu_5Au_1-TiO_2$, the 6-proton C_2H_6 is more accessible compared to 4-proton C_2H_4 .

Figure R3. ICP-AES patterns of $E_t-Cu_5Au_1-TiO_2$ and $H_t-Cu_5Au_1-TiO_2$.

[1] Feng, J. Q. et al. Modulating adsorbed hydrogen drives electrochemical CO_2 -to- C_2 products. *Nat. Commun.* **14**, 4615 (2023).

[2] Vasileff, A. et al. Surface and interface engineering in copper-based bimetallic materials for selective CO_2 electroreduction. *Chem* **4**, 1809 (2018).

[3] Zhang, L. et al. Redispersion of exsolved Cu nanoparticles on $LaFeO_3$ photocatalyst for tunable photocatalytic CO_2 reduction. *Chem. Eng. J.* **452**, 139273 (2023).

[4] Kim, D. Y. et al. Synergistic geometric and electronic effects for electrochemical reduction of carbon dioxide using gold-copper bimetallic nanoparticles. *Nat. Commun.* **5**, 4948 (2014).

[5] Jiang, D. L. et al. Synergistic integration of AuCu co-catalyst with oxygen vacancies on TiO_2 for efficient photocatalytic conversion of CO_2 to CH_4 . *ACS Appl. Mater. Interfaces* **13**, 46772–46782 (2021).

Q4. There are many C-based intermediates in the whole CO_2 photoreduction process.

Why the author considers the *CO is the predominate adsorbate, please give more evidence.

Response:

Thank you for your rational replenishment for our work. The rationalisation of *CO adsorbate could be analyzed in theoretical and experimental evidence. As shown in DFT calculation (**Figure R4a**), there is almost no energy input for *CO_2 , *COOH , and *CO formation, which implies rapid *CO formation due to these exothermic and spontaneous processes. Nevertheless, all the conversions from *CO to *CHO , *OCCO , and CO molecules in these models must overcome an enormous energy barrier, indicating the rate-determined significance of *CO conversion during the CO_2 reduction process. Therefore, *CO species could be the most promising residual intermediates due to the difficult conversion and desorption. Moreover, the indeed residual ^{13}CO species were detected by GC-MS after the $^{13}CO_2$ photoreduction tests over $Cu-TiO_2$, $E_7-Cu-TiO_2$, and $Cu_5Au_1-TiO_2$ (**Figure R4b-d**). The corresponding revision and discussion have been added in manuscript (line 542-547).

Figure R4. a, The free energy diagram of CO_2 reduction to CO , *CHO , and *OCCO over different modeled surfaces. GC-MS spectra of the possible residual products absorbed on $Cu-TiO_2$ (b), $E_7-Cu-TiO_2$ (c), and $Cu_5Au_1-TiO_2$ (d) after the 24 h

photoreduction $^{13}\text{CO}_2$.

Q5. It is suggested to perform the FTIR spectrum of $\text{E}_7\text{-CuAu-TiO}_2$ before and after CO_2 photoreduction to analyze the possible C-based residual surficial absorbates.

Response:

Thank you for your rigorous guidance. As shown in **Figure R5**, there is no special characteristic peaks of C-based residual surficial absorbates was observed in FTIR spectrum after CO_2 photoreduction over $\text{E}_7\text{-Cu}_5\text{Au}_1\text{-TiO}_2$, suggesting the inexistence of residual C-based intermediate. Therefore, the FTIR spectrum further verify the superior stability of such CuAu-DAs structure. The corresponding revision and discussion have been added in manuscript (line 465-469, Supplementary Fig. 46).

Figure R5. FTIR spectrum of $\text{E}_7\text{-Cu}_5\text{Au}_1\text{-TiO}_2$ before and after CO_2 photoreduction.

Q6. The methods of the XAFS data fitting should be added in the manuscript, which is significant reference.

Response:

Thank you for your accurate advice. Data reduction, data analysis, and EXAFS fitting were performed and analyzed with the Athena and Artemis programs of the Demeter data analysis packages [1] that utilizes the FEFF6 program [2] to fit the EXAFS data. The energy calibration of the sample was conducted through a standard Cu foil, which as a reference was simultaneously measured. A linear function was subtracted from the pre-edge region, then the edge jump was normalized using Athena software. The k^3 -weighted $\chi(k)$ data were Fourier transformed after applying a Kaiser-Bessel window

function ($\Delta k = 1.0$). For EXAFS modeling, the global amplitude EXAFS (CN , R , σ^2 and ΔE_0) were obtained by nonlinear fitting, with least-squares refinement, of the EXAFS equation to the Fourier-transformed data in R-space, using Artemis software, EXAFS of the Cu foil is fitted and the obtained amplitude reduction factor S_0^2 value (0.845) was set in the EXAFS analysis to determine the coordination numbers (CNs) in the Cu-O and Cu-Au scattering path in sample. The corresponding revision and discussion have been added in manuscript (line 689-699, References 103 and 104).

[1] Ravel, B. & Newville, M. ATHENA, ARTEMIS, HEPHAESTUS: data analysis for X-ray absorption spectroscopy using IFEFFIT. *J. Synchrotron Radiat.* **12**, 537 (2005).

[2] Zabinsky, S. I. et al. Multiple-scattering calculations of x-ray-absorption spectra. *Phys. Rev. B* **52**, 2995 (1995).

Q7. From the photocatalytic results, it can be obviously observed that the C_2H_4 is the main product for $E_1-C_5A_1$, however, CH_4 is the main product for C_5A_1 and $H_{11}-C_5A_1$. The authors must provide more experimental characterization and explanations to clarify this point. This will provide readers more useful information to design efficient and selective catalysts.

Response:

Thank you for your meticulous suggestion. The products category of CO_2 photoreduction is often related to different atomic structure and components. Compared to CuAu-DAs structure, both the Cu_5Au_1 and $H_{11}-Cu_5Au_1-TiO_2$ are composed of Au single atoms (SAs) modified Cu nanoclusters (NCs), indicating the similar structural unit, which displays the consistently predominated CH_4 product (**Figure R6a,b**). In **Figure R6c,d**, the rate-determined step (RDS) of CuAu-DAs and Au-SAs modified Cu-NCs are corresponded to $*CO$ coupling and $*CO$ protonation, respectively, which directly determines the categories of CO_2 photoreduction. Consequently, both the experimental and theoretical analysis verify that the Au-SAs modified Cu-NCs tend to C_1 intermediate protonation and CH_4 production, while the CuAu-DAs benefits for the C-C coupling and C_2H_4 production. The corresponding revision and discussion have been added in manuscript (line 361-371, Supplementary Fig. 27).

Figure R6. Photocatalytic CO₂ reduction performance over E_T-Cu₅Au₁-TiO₂ (a) and H_T-Cu₅Au₁-TiO₂ (b). c, The free energy diagram of CO₂ reduction to CO, *CHO, and *OCCO over different modeled surfaces. d, The most energetically favorable *OCCO adsorption configurations.

Reviewer #3 (Remarks to the Author):

In this manuscript, the authors claim to have synthesized Cu-Au dual atoms (DAs) on TiO₂ via a selective leaching approach and investigate their efficacy in photoconversion of CO₂. They conducted extensive studies but the results seem not to sufficiently support the conclusions and thoroughly explored the beneficial effects of Cu-Au DAs in the photocatalytic reduction of CO₂. Although the performance seems very promising the formation mechanism and identification of Cu-Au DAs is not convincing. Therefore, I don't think the manuscript can be accepted in Nature Communications.

Response to comments:

We sincerely appreciate you for the constructive and valuable comments about our work. According to your excellent suggestion, we have deeply understood your misgivings about our work, especially for the actual components about the CuAu composites. Here, we have replenished the related experimental and theoretical reference for the

elimination about all the structural concerns. Here, we sincerely wish that you can give us a new opportunity to reconsider this work in publication of Nature Communications.

Q1. Necessary characterization of commercial TiO₂ such as SEM etc. should be added for better comparison with Cu₅Au₁-TiO₂.

Response:

Thank you very much for your rigorous suggestion. SEM, TEM, and EDS analysis of TiO₂ have been replenished to further distinguish the difference between TiO₂ and Cu₅Au₁-TiO₂. As shown in **Figure R1** and **R2**, the commercial TiO₂ displays the irregular nanoparticle morphology with the ca. 13.6 nm of particle size, and the EDS mapping analysis also verifies the uniform Ti and O elements distribution. Compared to commercial TiO₂, the regular and larger size (ca. 31.0 nm) of Cu₅Au₁ alloy was observed in **Figure R3a**, and the compactly connected heterointerface was also observed between Cu₅Au₁ and TiO₂ (**Figure R3b**). Although the EDS analysis (**Figure R4**) displays the 2.95 of Cu/Au atomic ratio in the selected area, and the overlapping Cu and Au elements distribution were also observed in EDS mapping images (**Figure R3c-g**), there is merely existence of apparent Au lattice in such CuAu alloy (**Figure R3b**). This observed phenomenon also could serve as the powerful reference for the later issues. The corresponding revision and discussion have been added in manuscript (line 93-105, Supplementary Figs. 3 and 4).

Figure R1. **a**, SEM image of TiO₂. The corresponding EDS mapping images of Ti (**b**) and O (**c**) elements in (**a**). **d**, TEM image of TiO₂. **e**, HRTEM image and particle size distribution. **f**, HRTEM image of TiO₂. **g**, HAADF-STEM image of TiO₂. The corresponding EDS mapping images of Ti (**h**) and O (**i**) elements in (**g**).

Figure R2. **a**, HAADF-STEM image of TiO₂. **b**, EDS spectrum of the selected area in (**a**).

Figure R3. **a**, TEM image of $\text{Cu}_5\text{Au}_1\text{-TiO}_2$. **b**, HRTEM image of the selected area in **(a)**. **c**, HAADF-STEM image of $\text{Cu}_5\text{Au}_1\text{-TiO}_2$. The corresponding EDS mapping images of Au **(d)**, Cu **(e)**, Ti **(f)**, and O **(g)** elements in **(c)**.

Figure R4. **a**, HAADF-STEM image of $\text{Cu}_5\text{Au}_1\text{-TiO}_2$. **b**, EDS spectrum of the selected area in **(a)**.

Q2. The identification of Cu SAs and Au SAs from the STEM images is not convincing. The resolution of the images is too low to identify the SAs (e.g. Figure 1c,d), let alone to distinguish Au and Cu from the negligible difference in contrast and size.

Response:

Thank you for your rigorous guidance. Based on the undistinguished resolution for different elemental configuration in such low contrasting gray images, the gray images were converted into pseudo-color images to further improve the distinguishability

between different elements, and the corresponding HRTEM images were also replenished. As shown in AC-HAADF-STEM (**Figure R5a,b**) and HRTEM (**Figure R5c,d**) images of E₁-Cu₅Au₁-TiO₂, the slightly destroyed CuAu lattice and few Cu vacancies were observed, suggesting the partial breakage of such ordered CuAu alloy. The AC-HAADF-STEM (**Figure R5e,f**) images of E₃-Cu₅Au₁-TiO₂ further display the severely destroyed Cu₅Au₁ lattice. Moreover, the distinguished ordered lattice of TiO₂ and disordered lattice of Cu₅Au₁ were simultaneously observed (**Figure R5g,h**), which implied the successfully vectored etching of such Cu₅Au₁ alloy. After the constant etching of Cu₅Au₁, the ablative Cu₅Au₁ alloy and few CuAu-DAs were obtained (**Figure R5e,f**), further indicating that the constant loss of Cu atoms could promote the collapse of the Cu₅Au₁ alloy framework and the redistribution of adjacent Cu-Au atoms on the TiO₂ surface. The corresponding revision and discussion have been added in manuscript (line 106-108 and 112-115, Supplementary Figs. 5 and 6).

Figure R5. (a,c) AC-HAADF-STEM and HRTEM images of $E_1\text{-Cu}_5\text{Au}_1\text{-TiO}_2$ and (b,d) the corresponding intensity-profile images of pseudo-color with calibration bar. (e,g) AC-HAADF-STEM and HRTEM images of $E_3\text{-Cu}_5\text{Au}_1\text{-TiO}_2$ and (f,h) the corresponding intensity-profile images of pseudo-color with calibration bar.

Q3. In the $E_{11}\text{-Cu}_5\text{Au}_1\text{-TiO}_2$ sample, it is obvious from the ICP result that Cu is still present, therefore in Figure 1g CuSAs may also be present and Figure 1a is not correct.

Response:

Thank you for your careful checks. We apologize for our carelessness about the AC-HAADF-STEM analysis of $E_{11}\text{-Cu}_5\text{Au}_1\text{-TiO}_2$. Based on your suggestion, we have added the detailed intensity profile about AC-HAADF-STEM images of $E_{11}\text{-Cu}_5\text{Au}_1\text{-TiO}_2$ to systematically summarize the distribution of Cu/Au atoms. As shown in **Figure R6**, we have investigated total 27 of single atoms decorated on the surface of TiO_2 . According to the comparison of contrast and size between different single atoms, only a few Cu single atoms (5/27) were observed on $E_{11}\text{-Cu}_5\text{Au}_1\text{-TiO}_2$, which is matched well with the result of ICP-AES (**Figure R7**). Therefore, compared to the CuAu-DAs structure in 7 h vectored etching process, the prolonged 11 h vectored etching time of Cu could further promote the dissociation of the CuAu-DAs structure. The corresponding revision and discussion have been added in manuscript (line 122-126, Fig. 1a, Supplementary Fig. 8).

Figure R6. AC-HAADF-STEM images and intensity profile of $E_{11}\text{-Cu}_5\text{Au}_1\text{-TiO}_2$.

Figure R7. ICP-AES patterns of $E_t\text{-Cu}_5\text{Au}_1\text{-TiO}_2$.

Q4. In a typical leaching process, one might anticipate the initial leaching of Cu on the outer surface of Cu₅Au₁. Subsequently, it's expected that the surrounding Au atoms bonded with those Cu atoms would also be affected. However, based on the ICP results, the amount of Au appears to remain constant. This observation requires a clear and detailed demonstration.

Response:

Thank you for your careful consideration. During the synthesis of CuAu alloying, the Cu²⁺ and Au³⁺ was reduced by ascorbic acid to form Cu⁰ and Au⁰. Owing to the different redox capacity related standard electrode potential (E^0) between Fe³⁺ ($\text{Fe}^{3+} + e^- \rightarrow \text{Fe}^{2+}$, $E^0 = +0.77 \text{ V}$) and Cu⁰ ($\text{Cu}^{2+} + 2e^- \rightarrow \text{Cu}^0$, $E^0 = +0.34 \text{ V}$), the Fe³⁺ could spontaneously be reduced by Cu⁰ [1,2], the detailed reaction as the following formula: $\text{Fe}^{3+} + \text{Cu}^0 \rightarrow \text{Fe}^{2+} + \text{Cu}^{2+}$. However, due to the insufficient oxidized capacity of Fe³⁺ for Au⁰ ($\text{Au}^{3+} + 3e^- \rightarrow \text{Au}^0$, $E^0 = +1.52 \text{ V}$) oxidation, the Au⁰ could not be oxidized by Fe³⁺ and keep the zero valence during the whole etching process [3-5], which seriously restrains the dissolution of Au⁰ in the acid solution and promote the redistribution of Au elements on TiO₂. Therefore, the results of ICP-AES display the constant amounts of Au after different etching time. Moreover, the control-experiment for the verification of constant Au amounts were also conducted. The contents of Au was increased to form the Cu₁Au₁ alloy, and the 9 h and 11 h vectored etching were also taken on Cu₁Au₁-TiO₂. As shown in **Figure R8a-c**, the different destroyed degree of Cu₁Au₁ alloy were observed, and the few and negligible Cu single atoms were observed in **Figure R8b** and **8c**, respectively, while the Au lattice fringes was still existed rather than collapse due to the stable skeleton composed of large amounts of Au. Besides, ICP-AES also displays the constant amounts of Au contents and dramatically decreased Cu contents (**Figure R8d**), further indicating that such leaching process is merely directional to Cu elements dissociation. The corresponding revision and discussion have been added in manuscript (line 239-246, Supplementary Table 3, References 64).

Figure R8. HRTEM image of Cu₁Au₁-TiO₂ (a). AC-HAADF-STEM images of E₉-Cu₁Au₁-TiO₂ (b) and E₁₁-Cu₁Au₁-TiO₂ (c). (d) Table of ICP-AES results of E_i-Cu₁Au₁-TiO₂.

[1] Liu, K. et al. Facile path for copper recovery from waste printed circuit boards via mechanochemical approach. *J. Hazard. Mater.* **400**, 129638 (2022).

[2] Liu, X. et al. Rapid oxidation of acetaminophen with zero-valent copper induced hydrogen peroxide process in presence of ferric ion and chloride ion. *Chem. Eng. J.* **470**, 144202 (2023).

[3] Milazzo, G., Caroli, S. & Sharma, V. K. Tables of standard electrode potentials. *J. Electrochem. Soc.* **125**, 261C (1978).

[4] Bard, A. J., Parsons, R. & Jordan, J. Standard Potentials in Aqueous Solutions. 848 (1985).

[5] Haynes, W. M. Electrochemical Series. *CRC Handbook of Chemistry and Physics* 10 (2011).

Q5. XRD analysis seems inconclusive. i. In ref 62, the XRD peak of AuCu is close to Au because Au is the main component, but in this study, Cu is the main component. ii. The observation of Au peaks is an obvious sign for the existence of Au nanoparticles. iii. There are no Cu diffraction peaks in the XRD pattern of Cu-TiO₂ sample, and the authors attribute this to the ultrafine distribution of Cu, but there is no characterization

to support this.

Response:

Thank you for your accurate guidance and critical comment. The answers to question (i-iii) have been listed as follow:

(i) It has been reported that XRD pattern merely represented the reflections related to all even or Miller indexes [1,2]. Chen et al. reported the different ordering degrees of CuAu alloy (**Figure R9,10**), and the low ordering degree of CuAu alloy merely display the Au characteristic peaks due to the dominated Miller indexes of Au phase [1]. Although the AC-HAADF-STEM displays the Cu atoms surrounded isolated Au single atoms (**Figure R11a**), the predominated 0.233 nm of lattice spacing was observed in the AC-HAADF-STEM of $\text{Cu}_5\text{Au}_1\text{-TiO}_2$, which is ascribed to the (111) crystal facet of Au rather than Cu and CuAu characteristic lattice planes. Therefore, both the AC-HAADF-STEM and XRD reflections display the characteristic (111) crystal facet due to the dominated Miller indexes of Au phase (**Figure R11a-c**). Although the overlapping Cu and Au elements distribution were also observed in EDS mapping images (**Figure R12c-g**), and the EDS analysis (**Figure R13**) displays the 2.95 of Cu/Au atomic ratio in the selected area, there is merely existence of apparent Au lattice in such CuAu alloy (**Figure R12b**), which further confirms the dominated Miller indexes of Au phase. In addition, the same synthetic method about CuAu alloy was applied in C_3N_4 photocatalyst, and **Figure R14c** only displays the crystal facet of Au, while HAADF-STEM (**Figure R14d**) and the corresponding elemental mapping (**Figure R14e-h**) verify the superimposed Cu and Au elements distribution, indicating the existence of CuAu alloy. Although the EDS analysis (**Figure R15**) displays the 4.42 of Cu/Au atomic ratio, there is still no Cu lattice exhibited due to the nondominated Miller indexes of Cu phase. Therefore, such evidence about the lattice engineering could be valuable guidance for the investigation of detailed alloying-based structure. The corresponding revision and discussion have been added in manuscript (**line 219-228, Supplementary Figs. 4d-g and 13**).

(ii) As suggested by the referee, we have verified the existence of ordered CuAu alloy structure (isolated Au single atoms in Cu lattice). However, the small pure Au phase are

also indeed existence under such solvothermal-reduction approach due to the different reduction potentials of Au ($[\text{AuCl}_4]^-/\text{Au}^0$, $E^0 = +0.93$ V) and Cu ($\text{Cu}^{2+}/\text{Cu}^0$, $E^0 = +0.34$ V) [3-6], which contributes to the rapid growth of pure Au phase and the restrained ordered growth of Cu in the CuAu structure. Despite it is a disturbing issue at current, we believe that we must pick out an optimized synthetic method to construct the highly-ordered CuAu alloy in our future researches. Therefore, we still insist that such up-bottom synthetic strategy is potential for the flexible fabrication of atomic level catalysts even under some small challenges. The corresponding revision and discussion have been added in manuscript (line 208-214, Supplementary Fig. 13).

(iii) HRTEM and EDS analysis were taken to figure out the detailed configuration of pure Cu on TiO_2 surface (Figure R16a-d,17). As shown in HRTEM and EDS mapping images, the small-sized Cu nanoparticle (ca. 7.9 nm) were decorated on TiO_2 (Figure R16a-d), and the EDS mapping also exhibited the uniformly dispersed Cu elements, which indicated that Cu existed as the small-sized scale rather than extended growth into large-sized structure. In addition, when the same metal loading amounts of Au and Cu were loaded on TiO_2 , respectively, the apparent Au characteristic peaks were observed in XRD pattern (Figure R18), while no characteristic peaks of Cu were exhibited, indicating the uniformly dispersion and low crystal degree of Cu. After doubling the addition of Cu contents, there is still no Cu characteristic peaks were observed, further indicating that the small-sized uniformly dispersion of Cu is the dominated reason for the blank Cu characteristic peaks rather than the loading amounts. The corresponding revision and discussion have been added in manuscript (line 214-219, Supplementary Figs. 14 and 15).

Figure R9. **a**, XRD diffraction patterns of AuCu BNs with different ordering degrees, which are normalized by the intensity of (111) peak; the inset is the comparison of the intensity of the (110) peak of different AuCu BNs. **b**, The representative HAADF-STEM image of a 90 %-AuCu NP; scale bar = 2 nm. The inset is the proposed model of ordered AuCu, where the yellow atom is Au and the red atom is Cu. (Reference from *Angew. Chem. Int. Ed.* 2022, 61, e202117834).

Figure R10. The representative HAADF-STEM image of a (a) 0%-AuCu nanoparticle, (b) 30%-AuCu nanoparticle, c, 60%-AuCu nanoparticle. The scale bar is 2 nm. The inset is the proposed model of ordered AuCu, where the yellow atom is Au and the red atom is Cu. (Reference from *Angew. Chem. Int. Ed.* 2022, 61, e202117834)

Figure R11. AC-HAADF-STEM images (a) and intensity profile (b) of $\text{Cu}_5\text{Au}_1\text{-TiO}_2$. (c) XRD patterns of TiO_2 , Au-TiO_2 , Cu-TiO_2 , and $\text{Cu}_5\text{Au}_1\text{-TiO}_2$.

Figure R12. a, TEM image of $\text{Cu}_5\text{Au}_1\text{-TiO}_2$. b, HRTEM image of the selected area in (a). c, HAADF-STEM image of $\text{Cu}_5\text{Au}_1\text{-TiO}_2$. The corresponding EDS mapping images of Au (d), Cu (e), Ti (f), and O (g) elements in (c).

Figure R13. a, HAADF-STEM image of $\text{Cu}_5\text{Au}_1\text{-TiO}_2$. b, EDS spectrum of the selected area in (a).

Figure R14. a, TEM image of C_3N_4 . b, TEM image of $\text{Cu}_5\text{Au}_1\text{-C}_3\text{N}_4$. c, HRTEM image of $\text{Cu}_5\text{Au}_1\text{-C}_3\text{N}_4$ that magnified in (b). d, HAADF-STEM image of $\text{Cu}_5\text{Au}_1\text{-C}_3\text{N}_4$. e-h, The corresponding EDS mapping images of $\text{Cu}_5\text{Au}_1\text{-C}_3\text{N}_4$ in (d).

Figure R15. a, HAADF-STEM image of $\text{Cu}_5\text{Au}_1\text{-C}_3\text{N}_4$. b, EDS spectrum of the selected area in (a).

Figure R16. **a**, HRTEM image of Cu-TiO₂. **b**, Magnifying HRTEM image of Cu₅Au₁-C₃N₄ in **(a)**. Acquired HRTEM images intensity profile of **(c)** and **(d)** in **(b)**. **e**, HAADF-STEM image of Cu-TiO₂. The corresponding EDS mapping images of O **(f)**, Ti **(g)**, and **(h)** in **(e)**.

Figure R17. **a**, HAADF-STEM image of Cu-TiO₂. **b**, EDS spectrum of the selected area in **(a)**.

Figure R18. XRD pattern of TiO₂ modified with Au and different Cu loading amounts.

[1] Chen, H. Q. et al. Unmasking the critical role of the ordering degree of bimetallic nanocatalysts on oxygen reduction reaction by in Situ raman spectroscopy. *Angew. Chem. Int. Ed.* **61**, e202117834 (2022).

[2] Chen, H. Q. et al. The ordered and disordered nano-intermetallic AuCu/C catalysts for the oxygen reduction reaction: the differences of the electrochemical performance. *J. Electrochem. Soc.* **164**, F1654–F1661 (2017).

[3] Milazzo, G., Caroli, S. & Sharma, V. K. Tables of standard electrode potentials. *J. Electrochem. Soc.* **125**, 261C (1978).

[4] Bard, A. J., Parsons, R. & Jordan, J. Standard Potentials in Aqueous Solutions. 848 (1985).

[5] Chowdhury, I. H., Roy, M., Kundu, S. & Naskar, M. K. TiO₂ Hollow microspheres impregnated with biogenic gold nanoparticles for the efficient visible light-induced photodegradation of phenol. *J. Phys. Chem. Solids* **129**, 329–339 (2019).

[6] Chen, H. Q. et al. Unmasking the critical role of the ordering degree of bimetallic nanocatalysts on oxygen reduction reaction by in situ Raman spectroscopy. *Angew. Chem. Int. Ed.* **61**, e202117834 (2022).

Q6. The photocurrent spectra in Figure S12 look strange. The authors should provide some explanations.

Response: Thank your careful check and suggestion. The tremulous lines of photocurrent images could be ascribed to the loose adhesion between photocatalysts and FTO. Herein, we have adopt the coating agent of naphthol to replace the previous polyvinylpyrrolidone (PVP), and the exfoliation phenomenon of photocatalysts has been deeply restrained based on the compact connection between photocatalysts and FTO. Meanwhile, the gradients of photocurrent intensity in the optimized testing approach are the same as the previous testing approach (**Figure R19a**), which further suggests that the CuAu-DAs exhibits much higher charge separation efficiency and faster interfacial charge transportation than TiO_2 , Cu-TiO_2 , and $\text{Cu}_5\text{Au}_1\text{-TiO}_2$. The corresponding revision and discussion have been added in manuscript (**Supplementary Fig. 20c**).

Figure R19. **a**, Transient photocurrent response of as-prepared catalysts with the coating agent of naphthol. **b**, Transient photocurrent response of as-prepared catalysts with the coating agent of PVP.

Q7. Using UV-Vis spectra to determine the bandgap is not precise. And it is also not reasonable to determine the band gap of TiO_2 deposited with metals.

Response:

Thank you for your careful correction. The band gap of composites is unmeaning for the investigation of band structure. Here we analyze the band alignment of independent TiO_2 and CuAu before and after contact. The valence-band X-ray photoelectron spectroscopy (VB-XPS) was replenished to further analyze the Fermi level (E_f) of pure

TiO₂, which benefits for the verification of band alignment and charge transfer pathways between TiO₂ and CuAu metals. Based on the data of UV-vis DRS spectra, the plot of $(ah\nu)^{1/2}$ vs. $h\nu$ was also fitted to further evaluate the bandgap energies of the as-prepared samples, and the bandgap energy (E_g) of TiO₂ is 3.26 eV (**Figure R20a,b**). The Mott-Schottky show the flat band potential of TiO₂ was estimated to be -0.74 V vs. Ag/AgCl (**Figure R20c**), which was equal to -0.55 V vs. NHE ($E_{\text{NHE}} = E_{\text{Ag/AgCl}} + 0.197$ V). Based on the positive MS plot of TiO₂, TiO₂ was defined as n-type semiconductor, and the flat band potential is approximate 0.3 V below to the CB potential (E_{CB}) in n-type semiconductor [1,2], indicating the -0.85 V vs. NHE of E_{CB} potential [3,4]. Meanwhile, the VB potential (E_{VB}) of TiO₂ was further calculated to be 2.41 V. In **Figure R20d**, the VB-XPS spectra exhibited the 1.69 eV energy gap between the E_f and E_{VB} , and the E_f could be calculated to be 0.72 V. It has been reported that the work functions of metal Cu and Au correspond to 4.65 and 5.10 eV [5], respectively, and the E_f of metal Cu and Au could be calculated to be 0.15 and 0.60 V by the formula of E (vs. NHE, pH = 0) = -4.5 eV – E (vacuum level) [6], indicating that the electron could further transfer from Cu to the lower E_f of Au (**Figure R20e**). Based on the difference of E_f between Cu and Au, the more efficient photogenerated charge carrier separation could be realized on the CuAu modified TiO₂ compared to Cu modified TiO₂. Due to the larger E_f value of TiO₂ compared to CuAu metal, the electrons would transfer from CuAu to TiO₂ until their Fermi levels are aligned to the equilibrium state (**Figure R20e**), and the ohmic-junction could be further fabricated to provide the highspeed photogenerated charge carrier transfer pathways [7,8]. The corresponding revision and discussion have been added in manuscript (**Supplementary Fig. 18**).

Figure R20. **a**, UV-vis-NIR DRS spectra of as-prepared samples without etching. **b**, Plot of $(\alpha hv)^{1/2}$ versus (hv) for the bandgap energy of TiO_2 . **c**, Mott-Schottky plots of TiO_2 . **d**, The survey XPS spectrum of valence band spectrum of TiO_2 . **e**, Schematics illustrating the electronic band structure of TiO_2 before and after contact with CuAu .

[1] Shi, W. N. et al. Controllable synthesis of Cu_2O decorated WO_3 nanosheets with dominant (0 0 1) facets for photocatalytic CO_2 reduction under visible-light irradiation. *Appl. Catal., B* **243**, 236 (2019).

[2] Jo, W. K., Kumar, S., Eslava, S. & Tonda, S. Construction of $\text{Bi}_2\text{WO}_6/\text{RGO}/\text{g-C}_3\text{N}_4$ 2D/2D/2D hybrid Z-scheme heterojunctions with large interfacial contact area for efficient charge separation and high-performance photoreduction of CO_2 and H_2O into

solar fuels. *Appl. Catal., B* **239**, 586 (2018).

[3] Huang, H. W. et al. Anionic group self-doping as a promising strategy: band-gap engineering and multi-functional applications of high-performance CO_3^{2-} -doped $\text{Bi}_2\text{O}_2\text{CO}_3$. *ACS Catal.* **5**, 4094 (2015).

[4] Xiao, J. D. et al. Super synergy between photocatalysis and ozonation using bulk g- C_3N_4 as catalyst: A potential sunlight/ O_3 /g- C_3N_4 method for efficient water decontamination. *Appl. Catal., B* **181**, 420 (2016).

[5] Meng, A. Y., Zhang, L. Y., Cheng, B. & Yu, J. G. Dual cocatalysts in TiO_2 photocatalysis. *Adv. Mater.* **31**, 1807660 (2019).

[6] Cao, S. W. et al. 2D/2D heterojunction of ultrathin MXene/ Bi_2WO_6 nanosheets for improved photocatalytic CO_2 reduction. *Adv. Funct. Mater.* **28**, 1800136 (2018).

[7] Cui, X. et al. Low-temperature Ohmic contact to monolayer MoS_2 by van der Waals bonded Co/h-BN electrodes. *Nano Lett.* **17**, 4781 (2017).

[8] Wan, Y. M. et al. Conductive and stable magnesium oxide electron-selective contacts for efficient silicon solar cells. *Adv. Energy Mater.* **7**, 1601863 (2017).

Q8. In XANES spectra, the Au L-edge shows only similar spectra to Au^0 in contrast to the Cu K-edge in which the spectra is different from Cu^0 . Also, why is the Cu-Au peak in the Cu K-edge relatively smaller than the Au-Cu peak which is predominant in the Au L-edge?

Response:

Thank you for your good advice. As shown in question 5, the reason for the formation of pure Au phase has been concluded to the rapid Au crystal growth based on the more favorable reduction potential compared to Cu, which suggests the inevitable Au-Au contribution in the XANES and FT-EXAFS (**Figure R21d,f**). Although the pure Au phase is indeed existence, it cannot deny the co-existed CuAu-DAs structure based on the up-bottom synthetic strategy. Therefore, the Cu foil was used to calculate the standard amplitude reduction factor ($S_0^2 = 0.845$, **Table R1**), and the Cu K-edge EXAFS analysis of $\text{E}_7\text{-Cu}_5\text{Au}_1\text{-TiO}_2$ in R spaces was exhibited in **Figure R22**. The EXAFS spectrum of $\text{E}_7\text{-Cu}_5\text{Au}_1\text{-TiO}_2$ was analyzed by using two backscattering paths (Cu-O

and Cu-Au). The best-fitting results exhibited that the coordination number of the O and Au in the first coordination sphere of $E_7\text{-Cu}_5\text{Au}_1\text{-TiO}_2$ is fitted to be ≈ 3.3 and ≈ 1.2 at distances of 1.93 and 2.91 Å, respectively, implying the Cu-SAs is merely adjacent with single Au atom. Therefore, there is no doubt that the existence of CuAu-DAs structure under such vectored etching process, while it is still recognized indeed small existence of pure Au phase due to the shortage of such solvothermal method for the fabrication of highly-ordered CuAu alloy. The corresponding revision and discussion have been added in manuscript (line 153-163, Supplementary Fig. 10 and Table 2).

Figure R21. XANES analysis of $E_7\text{-Cu}_5\text{Au}_1\text{-TiO}_2$ and reference samples at the Cu K-edge (a) and Au L-edge (b). Corresponding k^3 -weighted FT-EXAFS spectra in the R space for $E_7\text{-Cu}_5\text{Au}_1\text{-TiO}_2$ and references at the Cu K-edge (c) and Au L-edge (d). e, Cu K-edge WT-EXAFS spectra of Cu foil and $E_7\text{-Cu}_5\text{Au}_1\text{-TiO}_2$. f, Au L-edge WT-EXAFS spectra of Au foil and $E_7\text{-Cu}_5\text{Au}_1\text{-TiO}_2$.

Figure R22. Cu K-edge EXAFS analysis of E₇-Cu₅Au₁-TiO₂ in R spaces.

Sample	Shell	CN ^a	R(Å) ^b	σ ² (×10 ⁻³ Å ²) ^c	ΔE ₀ (eV) ^d	R factor
Cu foil	Cu-Cu	12*	2.54	8.5	4.0	0.003
Sample	Cu-O	3.3	1.93	4.8	5.2	0.01
	Cu-Au	1.2	2.91	7.0	9.8	

Table R1. EXAFS fitting parameters at the Cu K-edge for various samples ($S_0^2=0.845$). ^aCN, coordination number; ^bR, distance between absorber and backscatter atoms; ^cσ², Debye-Waller factor to account for both thermal and structural disorders; ^dΔE₀, inner potential correction; R factor indicates the goodness of the fit. S_0^2 was fixed to 0.845, according to the experimental EXAFS fit of Cu foil by fixing CN as the known crystallographic value. Fitting range: $2.5 \leq k$ (1/Å) ≤ 12.5 and $1.0 \leq R$ (Å) ≤ 3.0 (Cu foil); $2.0 \leq k$ (1/Å) ≤ 12 and $1.0 \leq R$ (Å) ≤ 3.0 (E₇-Cu₅Au₁-TiO₂). A reasonable range of EXAFS fitting parameters: $0.700 < S_0^2 < 1.000$; $CN > 0$; $0.02 > \sigma^2 > 0.003 \text{ \AA}^2$; $|\Delta E_0| < 10 \text{ eV}$; R factor < 0.02 .

Q9. In PL spectroscopy, Cu₅Au₁-TiO₂ shows similar lifetime to Cu-TiO₂. It is not clear that Au addition improves charge transfer efficiency.

Response:

Thank you for your considerate check and guidance. To figure out the efficiency of charge transfer, the theoretical band alignment and actual photoelectrochemical experiments were combined to further address this issue. The work functions of metal Cu and Au correspond to 4.65 and 5.10 eV [1], respectively, and the Fermi level of

metal Cu and Au could be calculated to 0.15 and 0.60 V by the formula of E (vs. NHE, $\text{pH} = 0$) = $-4.5 \text{ eV} - E$ (vacuum level) [2]. Based on the introduction of lower Fermi level Au in Cu, the electron could transfer from Cu to Au based on the different Fermi level, which can further improve the photogenerated charge transfer efficiency and restrain the charge recombination (**Figure R23a**). Due to the larger Fermi level value of TiO_2 compared to CuAu metal, the electrons would transfer from CuAu to TiO_2 until their Fermi levels are aligned to the equilibrium state, and the ohmic-junction could be further fabricated to provide the highspeed photogenerated charge carrier transfer pathways [3,4]. After contact between TiO_2 and CuAu, the photogenerated electron arose from TiO_2 may also vectorially transfer from Cu to Au after acrossing the ohmic-junction, indicating that the ternary heterostructure can further promote the photogenerated charge transfer efficiency. Although the little decreased PL lifetime of $\text{Cu}_5\text{Au}_1\text{-TiO}_2$ compared to Cu-TiO_2 is insufficient to clarify the improvement of photogenerated charge transfer efficiency, the apparent increased photocurrent intensity and shortened arc radius of $\text{Cu}_5\text{Au}_1\text{-TiO}_2$ can further repressnt the deep optimization of photogenerated charge transfer efficiency based on the introduction of Au (**Figure R23b,c**). The corresponding revision and discussion have been added in manuscript (line 274-282, Supplementary Figs. 18 and 20c,d).

Figure R23. **a**, Schematics illustrating the electronic band structure of TiO_2 before and after contact with CuAu. **b**, Transient photocurrent response of $Cu-TiO_2$ and $Cu_5Au_1-TiO_2$. **c**, EIS of $Cu-TiO_2$ and $Cu_5Au_1-TiO_2$.

[1] Meng, A. Y., Zhang, L. Y., Cheng, B. & Yu, J. G. Dual cocatalysts in TiO_2 photocatalysis. *Adv. Mater.* **31**, 1807660 (2019).

[2] Cao, S. W. et al. 2D/2D heterojunction of ultrathin MXene/ Bi_2WO_6 nanosheets for improved photocatalytic CO_2 reduction. *Adv. Funct. Mater.* **28**, 1800136 (2018).

[3] Cui, X. et al. Low-temperature Ohmic contact to monolayer MoS_2 by van der Waals bonded Co/h-BN electrodes. *Nano Lett.* **17**, 4781 (2017).

[4] Wan, Y. M. et al. Conductive and stable magnesium oxide electron-selective contacts for efficient silicon solar cells. *Adv. Energy Mater.* **7**, 1601863 (2017).

REVIEWER COMMENTS

Reviewer #1 (Remarks to the Author):

Decision: Accept

The authors have carefully addressed all the comments and improved the manuscript quality significantly, I recommend acceptance for publication in Nature Communications as is.

Reviewer #2 (Remarks to the Author):

The authors have carefully considered and revised reviewers' suggestions. I would like to recommend to accept this manuscript in its present form.

Reviewer #3 (Remarks to the Author):

The authors have improved the manuscript and some of the comments have been properly answered. However, there are still some issues that need to be addressed.

1. I remain skeptical about the authors' methodology for distinguishing between Cu and Au atoms. The authors rely on the intensity profile from STEM images, i.e. in Fig. S7 the width of the peak and in Fig. S8 (seems to be) the intensity of the peak. However, it's important to note that these variations can be caused by several factors and are very imprecise at such a low resolution.
2. The authors observed the presence of Au nanoparticles or a pure phase Au in the CuAu alloy, and in particular, the Au content remains constant during the etching process. This implies that the Au nanoparticles persist in the final structure obtained. Consequently, the catalytic mechanism goes beyond the CuAu diatom and includes the significant involvement of Au nanoparticles. The authors should demonstrate an approach to show that the remaining Au nanoparticles don't affect the performance.
3. The authors should carefully check their manuscript and figures for various mistakes. E.g. "C3N4" in the figure caption of Figure S14; "XAFS date analysis"

Reviewer #1 (Remarks to the Author):

Decision: Accept

The authors have carefully addressed all the comments and improved the manuscript quality significantly, I recommend acceptance for publication in Nature Communications as is.

Response to remarks: Thank you for your careful review about our work. We appreciate the reviewer for recognizing our revision and the recommendation for publication.

Reviewer #2 (Remarks to the Author):

The authors have carefully considered and revised reviewers' suggestions. I would like to recommend to accept this manuscript in its present form.

Response to remarks: We appreciate the reviewers for your effort in carefully reviewing our manuscript and providing valuable feedback. Thank you very much for your recognition about our work.

Reviewer #3 (Remarks to the Author):

The authors have improved the manuscript and some of the comments have been properly answered. However, there are still some issues that need to be addressed.

Response to remarks: We sincerely thank you for your thorough evaluation and examination of our work. We also replenish the necessary evidence to enrich our research and contribute the more precise manuscript.

Q1. I remain skeptical about the authors' methodology for distinguishing between Cu and Au atoms. The authors rely on the intensity profile from STEM images, i.e. in Fig. S7 the width of the peak and in Fig. S8 (seems to be) the intensity of the peak. However, it's important to note that these variations can be caused by several factors and are very imprecise at such a low resolution.

Response: Thank you for your rigorous and important suggestion. The intensities of HAADF-STEM images are recognized as the predominated index for the identification of different atomic-level information due to the Z-contrast difference between different

atoms [1-3], and the HAADF-STEM intensities of the bright spots is related to variation of atomic mass of the corresponding elements [4,5], which implies that the comparison of the bright spots intensities is a substantial approach to distinguish different single atoms, especially for the obvious mass difference. Following by the suggestion of reviewer and references, we found the apparent mass difference between Cu ($m = 63.6$) and Au ($m = 197.0$), thus we revise the format of HAADF-STEM images into the comparison of peak intensity to further distinguish the different configuration between Cu and Au at atomic-level (Figure R1 and R2). The corresponding revision and discussion have been added in manuscript (line 117-122, Supplementary Figs. 7 and 8).

Figure R1. Acquired AC-HAADF-STEM images intensity profile of E7-Cu₅Au₁-TiO₂.

Figure R2. AC-HAADF-STEM images and intensity profile of $E_{11}\text{-Cu}_5\text{Au}_1\text{-TiO}_2$.

[1] Liu, L. Z. et al. One-dimensional single atom arrays on ferroelectric nanosheets for enhanced CO_2 photoreduction. *Nat. Commun.* **15**, 305 (2024).

[2] Liu, W. et al. Highly-efficient RuNi single-atom alloy catalysts toward chemoselective hydrogenation of nitroarenes. *Nat. Commun.* **13**, 3188 (2022).

[3] Liu C. et al. Catalytic activity enhancement on alcohol dehydrogenation via directing reaction pathways from single- to double-atom catalysis. *J. Am. Chem. Soc.* **144**, 4913-4924 (2022).

[4] Wang C. et al. Co and Pt dual-single-atoms with oxygen-coordinated Co–O–Pt dimer sites for ultrahigh photocatalytic hydrogen evolution efficiency. *Adv. Mater.* **33**, 2003327 (2021).

[5] Huang F. et al. Low-temperature acetylene semi-hydrogenation over the $\text{Pd}_1\text{-Cu}_1$

dual-atom catalyst. *J. Am. Chem. Soc.* **144**, 18485-18493 (2022).

Q2. The authors observed the presence of Au nanoparticles or a pure phase Au in the CuAu alloy, and in particular, the Au content remains constant during the etching process. This implies that the Au nanoparticles persist in the final structure obtained. Consequently, the catalytic mechanism goes beyond the CuAu diatom and includes the significant involvement of Au nanoparticles. The authors should demonstrate an approach to show that the remaining Au nanoparticles don't affect the performance.

Response:

Thank you for your crucial and considerate suggestion about the photocatalytic mechanism. The photocatalytic tests and theoretical analysis are conducted on different CuAu related models. In **Figure R3a**, only little of CO and CH₄ production are observed on pure Au nanoparticles (NPs) modified TiO₂ (Au-TiO₂) during the CO₂ reduction process, indicating that the pure Au NPs could not favor the C₂ products evolution. When few Cu species attached on Au NPs (Cu₁Au₅-TiO₂), the CH₄ production increases hugely compared to pure Au NPs, while almost no C₂ products are detected, implying that the synergistic effects between adjacent Cu atoms and Au NPs merely can promote the protonation of C₁ intermediates rather than C-C coupling reaction. Moreover, the E₇-Cu₅Au₁-TiO₂ displays the dramatically increased C₂H₄ and C₂H₆ activity and selectivity, which uncovers the predominated C-C coupling reaction on such CuAu diatomics (DAs) structure. The density functional theory (DFT) simulations of different CuAu structures are replenished to distinguish the relationship between structure and performance, especially for the crucial step of *CO conversion. As shown in **Figure R3b**, during the possible routes of *CO conversion on pure Au nanoclusters (NCs) modified TiO₂, the CO desorption is the most energetically favorable, indicating the predominated C₁ products evolution on pure Au modified TiO₂. After the Cu single atoms (SAs) introduction in Au NCs (**Figure R3c**), the protonation of *CO is more energetically favorable on CuAu-SANCs-TiO₂ compared to the dominated CO desorption on Au-NCs-TiO₂, which further replenishes that the synergistic effects between Cu SAs and Au NCs could not promote the C-C coupling

in thermodynamics. Notably, only the combination of Cu SAs and Au SAs (CuAu-DAs) could enormously decrease the C-C coupling energy barrier and turn the C-C coupling into the predominated route during the CO₂ reduction process (**Figure R3d**), which demonstrates that the C₂ products are mainly ascribed to the thermodynamic advantage of C-C coupling on CuAu-DAs structure. The corresponding revision and discussion have been added in manuscript (**Supplementary Fig. 49**).

Figure R3. a, CH₄, C₂H₄, C₂H₆, CO, and H₂ production rates of Au-TiO₂, Cu₁Au₅-TiO₂, and E₇-Cu₅Au₁-TiO₂. The free energy diagram of CO₂ reduction to CO, *CHO, and *OCCO over (b) Au-NCs-TiO₂, (c) CuAu-SANCs-TiO₂, and (d) CuAu-DAs-TiO₂ modeled surfaces.

Q3. The authors should carefully check their manuscript and figures for various mistakes. E.g. “C3N4” in the figure caption of Figure S14; “XAFS date analysis”

Response:

Thank you for your valuable advice. We have checked and revised this manuscript in detail to avoid such mistakes.